Brief Communication

# Open-3DSIM: an open-source three-dimensional structured illumination microscopy reconstruction platform

Ruijie Cao [1,2], Yaning Li [1,2], Xin Chen[1,2], Xichuan Ge[3], Meiqi Li [1,2], Meiling Guan[1,2], Yiwei Hou [1,2], Yunzhe Fu[1,2], Xinzhu Xu[1,2], Christophe Leterrier[4], Shan Jiang[5], Baoxiang Gao[3] & Peng Xi [1,2] ✉

Open-3DSIM is an open-source reconstruction platform for three-dimensional structured illumination microscopy. We demonstrate its superior performance for artifact suppression and high-fidelity reconstruction relative to other algorithms on various specimens and over a range of signal-to-noise levels. Open-3DSIM also offers the capacity to extract dipole orientation, paving a new avenue for interpreting subcellular structures in six dimensions ($xyz\theta\lambda t$). The platform is available as MATLAB code, a Fiji plugin and an Exe application to maximize user-friendliness.

Structured illumination microscopy (SIM) is the most universally implemented super-resolution modality in the life sciences because it offers fast longitudinal imaging with low phototoxicity and is highly compatible with fluorescent labeling[1–3]. With the flourishing of SIM, a variety of open-source reconstruction algorithms have been developed, such as OpenSIM[4], fairSIM[5], SIMtoolbox[6], HiFi-SIM[7] and so on. The availability of open-source software also boosts custom-built SIM hardware platforms, such as SLM-SIM[8], DMD-SIM[9], galvanometer-SIM[10], Hessian-SIM[11] and so on. Combining software and hardware has created an open and productive community for SIM researchers.

Compared with 2DSIM, 3DSIM doubles resolution along the $z$ axis[1,12–14] as well as in the $xoy$ plane. 3DSIM reconstruction algorithms can be found in commercial systems such as GE OMX and Nikon N-SIM, or open-source software such as Cudasirecon[1], AO-3DSIM[14], SIMnoise[15] and 4BSIM[16]. However, the commercial solutions are limited to specific microscopy platforms. The open-source solutions are all target-specific tools to solve certain imaging problems and are unsuitable for generic 3DSIM reconstruction. They may also lead to serious artifacts or offer poor user-friendliness. On the contrary, in the field of 2DSIM or single-layer 3DSIM, OpenSIM[4] explains the principle of SIM reconstruction systematically; fairSIM[5] integrates the algorithm into Fiji to facilitate use by biological researchers; HiFi-SIM[7] notably optimizes reconstruction results and has a user-friendly graphical user interface.

The lack of convenient multilayer 3DSIM software impedes users from accessing and using it, and serious artifacts challenge the fidelity and reliability of 3DSIM. Therefore, a well-established and user-friendly 3DSIM reconstruction tool is urgently needed in the 3DSIM field to ensure its further development.

To address this need, here we report Open-3DSIM, which can provide superior and robust multilayer 3DSIM reconstruction. We prepare the Fiji version to make it easily accessible for biological users; provide intermediate results to help hardware specialists to check their home-built 3DSIM data and open modular source codes for software developers to boost its future developments. Through comparisons with different algorithms on various specimens and signal-to-noise (SNR) levels, we demonstrate that Open-3DSIM offers superior performance due to the optimization of parameter estimation and spectral filtering, resulting in high-fidelity reconstructions with minimal-artifacts and preserved weak information. Furthermore, Open-3DSIM can extract the inherent dipole orientation information, unlocking the full potential of 3DSIM in multilayer, multicolor, polarization and time-lapse super-resolution reconstruction.

The principle of Open-3DSIM is shown in Supplementary Fig. 1 and Supplementary Note 1. The pattern of three-dimensional (3D) structured illumination[17] is generated by the interference of three beams through grating diffraction. Then, 3D stack data are taken

[1]Department of Biomedical Engineering, College of Future Technology, Peking University, Beijing, China. [2]National Biomedical Imaging Center, Peking University, Beijing, China. [3]Key Laboratory of Analytical Science and Technology of Hebei Province, College of Chemistry and Environment Science, Hebei University, Baoding, China. [4]Aix-Marseille Université, CNRS, INP UMR7051, NeuroCyto, Marseille, France. [5]Institute of Biomedical Engineering, Beijing Institute of Collaborative Innovation, Beijing, China. ✉e-mail: xipeng@pku.edu.cn

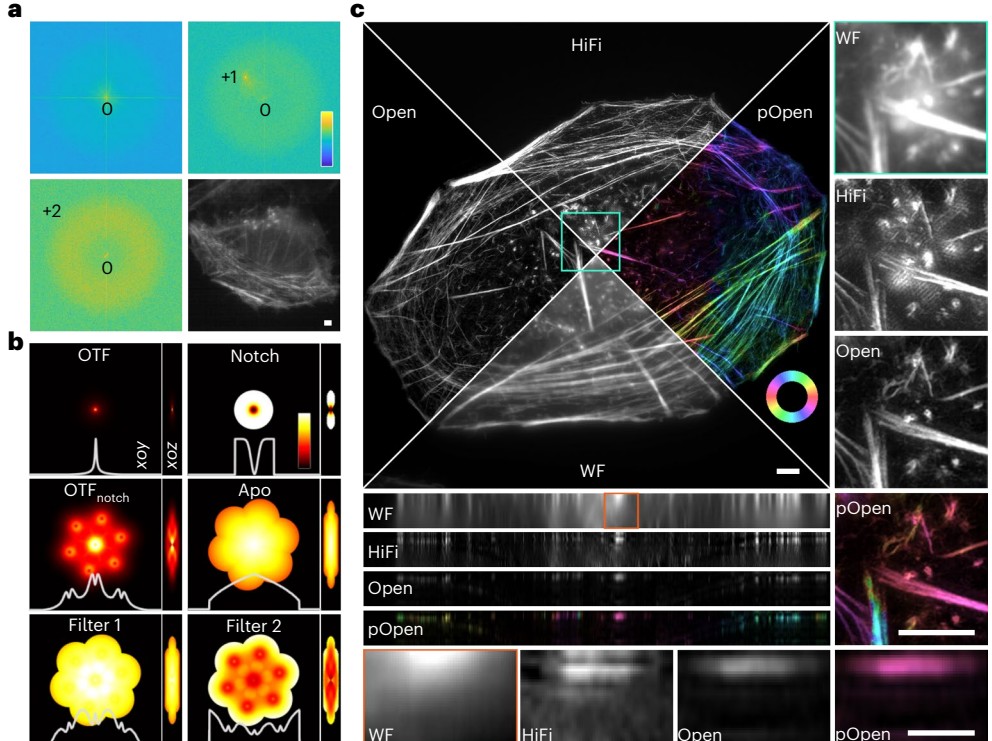

**Fig. 1 | Principle and reconstruction results of Open-3DSIM. a**, Separated frequency domain and image under low SNR, demonstrating the first frequency peak is higher than the second and is easier to recognize. The top left, top right and bottom left are the separated zeroth, first and second frequency domains. The bottom right is the raw image of actin filaments in U2OS under extremely low SNR, but the +first pattern (visible because of relatively low frequency) is easy to recognize. The color bars in the upper-right corner of **a** are in the format 'Intensity (a.u.)'. **b**, Two-step filter designed by OTF, Notch, $OTF_{notch}$ and Apo to optimize the frequency spectrum of 3DSIM, achieving the effect of reducing artifacts and improving resolution. The white lines below represent their corresponding profiles along the estimated frequency. **c**, Reconstruction results of actin filaments in U2OS, including the comparison of the single-layer algorithm (HiFi-SIM), the multilayer algorithm (Open-3DSIM), wide-field (WF) image and polarized Open-3DSIM image (pOpen). The color bars in 'pOpen' of **c** are the format 'Angle (rad)'. Scale bar, 2 μm. Scale on the *z* axis, 12 layers, 0.125 μm per layer. The experiments were repeated three times independently with similar results.

layer by layer, converted to the frequency domain and separated by a phase-separation matrix. The separated ±first frequency components are shifted to fill the leaky cone of zero frequency component, and the ±second frequency components are shifted to expand the spectrum range of the *xoy* plane. So, 3DSIM doubles the spectral range compared with wide field by filling the 'missing cone' in an optical conversion function (OTF) (Extended Data Fig. 1 and Supplementary Notes 1 and 2), and obtaining 3D super-resolution results.

We use the cross-correlation method[5,7] to estimate the frequency, angle, phase and modulation depth of the structured illumination patterns to avoid the involvement of initial parameters that other 3DSIM algorithms require to input[1,14–16]. To obtain a correct frequency parameter, which is the first step of parameter estimation, we use both +first and +second frequency components. Although traditionally estimating the peak of the +second frequency is more accurate than +first, the +first frequency's peak carries a higher contrast than the +first frequency in low SNR as shown in Fig. 1a. Therefore, we set up a criterion to determine whether estimating through the +second frequency's peak is reliable. If not, we will use the +first spectrum to estimate instead to guarantee the validity of frequency estimation. Then the corresponding parameter such as phase and angle can be accurately resolved. This method can greatly improve the correctness of parameter estimation when reconstructing images under low SNR, thus reducing various artifacts caused by parameter estimation errors[17,18].

Next, inspired by Hifi-SIM[7], we designed a two-step filter in the frequency domain based on the notch function (Notch), apodization function (Apo), optical conversion function (OTF) and $OTF_{notch}$

according to the estimated frequency vector in the *xoy* and *yoz* plane. First, we designed a notchfilter corresponding to the estimated frequency and input image size to improve the compatibility of different images. We demonstrate that the cooperation of Filter 1 and Filter 2 will make the 3D spectrum approach the ideal spectrum, being smooth and even[18,19], which can suppress the spectrum's peak, reduce the noise and compensate for high-frequency blurring (Fig. 1b, Extended Data Fig. 1 and Supplementary Note 1 and 2). This method to correct an abnormal spectrum is important for controlling the artifacts of reconstructed images and further improving the 3D resolution, making it perform well under low SNR. We carefully illustrate the function and superiority of the spectrum filters (Extended Data Fig. 2 and Supplementary Note 3) and provide a guide for adjusting the parameters (Extended Data Fig. 3 and Supplementary Note 4).

Figure 1c shows the 3DSIM imaging of actin filaments in U2OS. Open-3DSIM can improve *xyz* resolution compared to the wide-field images. What is more, Open-3DSIM interpreting whole actin structure can greatly eliminate defocus and artifacts with *z* axis improvement compared with single-layer reconstruction just like HiFi-SIM (Extended Data Fig. 4, Supplementary Note 5). We also introduce polarization dimension on the *xoy* plane to realize dipole orientation imaging in Open-3DSIM[20,21]. The dipole orientation information of the actin filament analyzed is shown in Fig. 1c. The parallelism between dipole orientation and actin filament direction validates the correctness of the polarization information.

To further test the performance of Open-3DSIM, we first conduct the simulation using a petal-shaped structure and a resolution

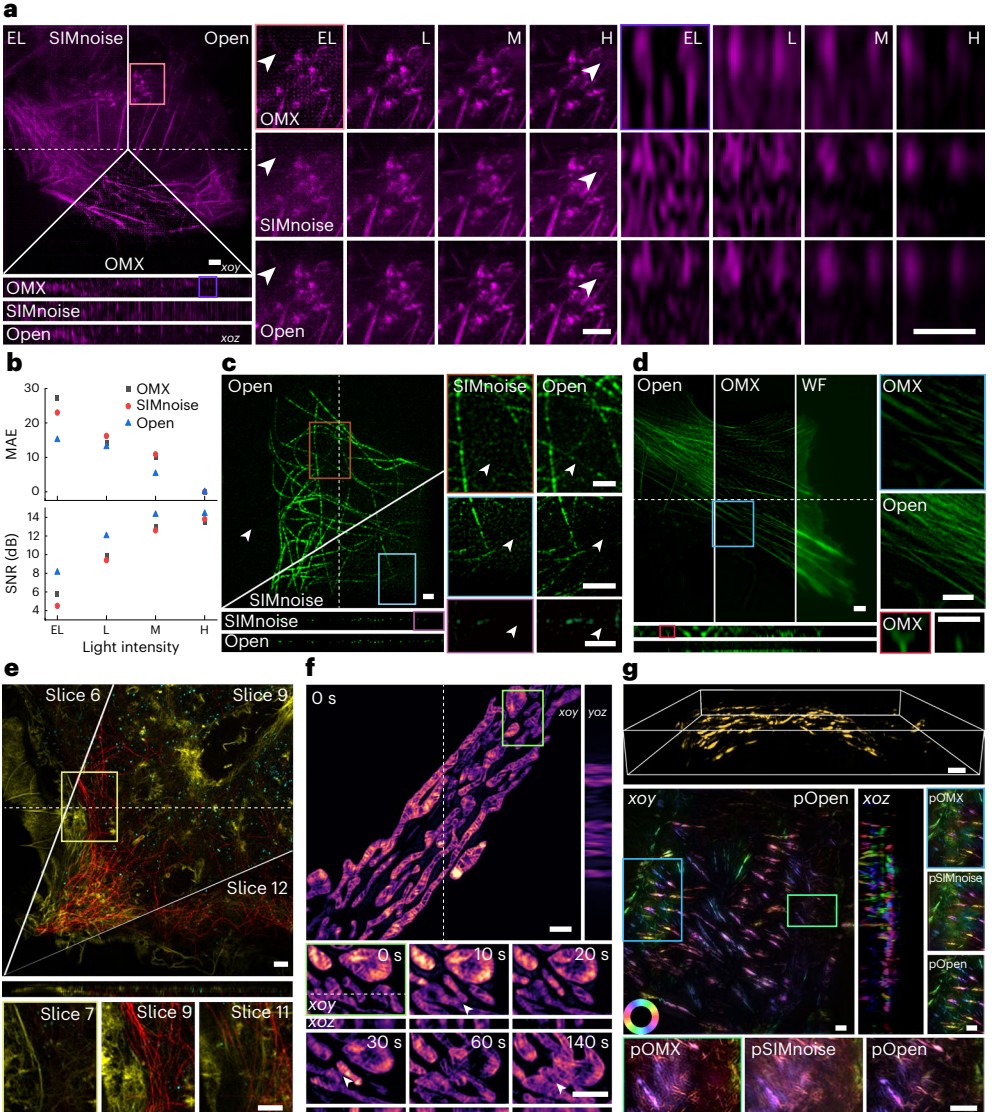

**Fig. 2 | Reconstruction performance of Open-3DSIM. a**, Reconstruction for actin filaments under different illumination conditions of extremely low (EL, intensity of 2%, exposure time of 5 ms), low (L, 5%, 5 ms), moderate (M, 10%, 20 ms) and high (H, 10%, 50 ms) with the comparison among SIMnoise, OMX and Open-3DSIM. **b**, MAE under different SNRs and algorithms, taking the corresponding highest SNR results as ground-truth images, with the relative SNR of the reconstruction results under different SNRs and algorithms to demonstrate the algorithm's ability to suppress a defocused background. **c**, State-of-the-art reconstruction results of tubulin structure from SIMnoise under the lowest SNR with the corresponding comparison. **d**, Reconstruction for actin filaments in U2OS cell from OMX under a low SNR (5%, 20 ms) with the corresponding comparison. **e**, The multi-color reconstruction by Open-3DSIM

of a COS cell labeled for actin filaments (yellow), clathrin (cyan), and tubulin (red), excited at 488, 561 and 640 nm in wavelength, respectively. **f**, Open-3DSIM analyzed the 3D mitochondrial ridge structure in live COS7 cell and showed the process of mitochondrial fusion, separation and apoptosis under a low SNR (10%, 3 ms) for 15 frames (10 s and six slices per frame). **g**, Reconstruction and polarization analysis of actin filaments in mouse kidney sections, including whole 3D sight (on the top) and the maximum-intensity projection on the *xoy* and *xoz* planes, with the comparison between pOMX, pSIMnoise and pOpen. The color bars in **g** are in the format 'Angle (rad)'. Scale bar is 2 μm. Scales on the *z* axis are: **a**, 13 layers; **c**, 41 layers; **d**, ten layers; **e**, 13 layers; **f**, six layers; and **g**, 41 layers. **a**, **c**, **d**, **f**, **g** 0.125 μm per layer; **e**, 0.12 μm per layer.

test 3D structure, gaining high-fidelity results with no observable artifacts (Extended Data Fig. 5 and Supplementary Note 6). We also compare Open-3DSIM with traditional Wenier-based 3DSIM reconstruction algorithms such as AO-3DSIM and 4BSIM, finding Open-3DSIM outperforms them compared with their well-adjusted reconstructed results (Extended Data Fig. 6 and Supplementary Note 7). Because SIMnoise optimizes the Wenier-based filter, we compare the reconstruction results of Open-3DSIM, OMX and SIMnoise under gradient SNR for actin filaments in U2OS in Fig. 2a. Here, Open-3DSIM outperforms OMX and SIMnoise in different levels of SNR with fewer artifacts and backgrounds. Also, the mean absolute

error (MAE) and SNR of Open-3DSIM are almost the same as those of other algorithms with higher illumination conditions in Fig. 2b, proving the excellent performance of Open-3DSIM under low SNR. Although SIMnoise has optimized the reconstruction under low SNR, Open-3DSIM shows better edge information retention and a denoising effect compared with the state-of-the-art result of SIMnoise in Fig. 2c with less reconstruction time. Similarly, compared with OMX, Open-3DSIM has an excellent ability to remove artifacts and retain weak information in Fig. 2d. More comparisons of reconstruction on the fluorescent quantitative standard sheet (Argolight) and various biological samples can be seen in Extended Data Figs. 7 and

8 and Supplementary Notes 8 and 9 to further prove the excellent performance of Open-3DSIM.

Furthermore, we demonstrate that Open-3DSIM can accurately reconstruct multicolor samples in Fig. 2e with N-SIM system and Extended Data Fig. 9, and Supplementary Note 10 with the OMX system, which shows excellent optical sectioning ability and compatibility with different microscope workstations. Open-3DSIM is also applied to analyze the 3D mitochondrial ridge structure in time-lapse imaging. We observe a clear ridge structure with the process of mitochondrial fusion, separation and apoptosis under low SNR as shown in Fig. 2f. Last, we use Open-3DSIM to reconstruct actin filament structure in a mouse kidney with dipole orientation imaging shown in Fig. 2g; it can be seen that polarized Open-3DSIM image (pOpen) outperforms polarized OMX image (pOMX) and polarized SIMnoise image (pSIMnoise) to a great extent in artifact suppression and defocus elimination, and more reconstructions of fluorescent dipole orientation are shown in Extended Data Fig. 10 (Supplementary Note 11). It is noteworthy that, with intensity calibration, the dipole orientation will be more accurately resolved by Open-3DSIM than with default assumptions. To help users to use Open-3DSIM, we also provide the parameters of reconstruction in Supplementary Table 1 and typical 3DSIM data in Supplementary Table 2 with results, parameters and comparisons.

With excellent performance on high-fidelity, low-artifact, defocus-removal and weak information retaining, Open-3DSIM can truly unleash the potential of 3DSIM in lateral and axial super-resolution, through multicolor, multilayer, time-lapse and dipole orientation imaging. Open-3DSIM is multiplatform, not limited by specific microscope workstations and can be customized and adjusted according to user needs. It is also extensible, as it is fully compatible with other algorithm optimization methods based on regular terms or machine learning. We expect Open-3DSIM to be the open-source standard for the 3DSIM multi-dimensional reconstruction and believe that open-source efforts will make a great difference to the community.

## Online content

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

## Methods

### Sample preparation

Human osteosarcoma U2OS cell lines (HTB-96, ATCC) were cultured in Dulbecco's modified Eagle's medium (DMEM, GIBCO) containing 10% heat-inactivated fetal bovine serum (GIBCO) and 100 U ml$^{-1}$ penicillin and 100 µg ml$^{-1}$ streptomycin solution (PS, GIBCO) at 37 °C in an incubator with 95% humidity and 5% $CO_2$. Fixed cells were plated on no. 1.5H cover glasses (CG15XH, Thorlabs) before they were grown to a suitable density (24 h) and fixed with 4% formaldehyde (R37814, Invitrogen) for 15 min. Then, the cells were washed three times with PBS to remove the formaldehyde. Alexa Fluor 488 Phalloidin (A12379, Invitrogen) was used to stain the actin filaments for 1 h at room temperature. Then, the coverslip was sealed on the slide with the prolong antifade mountant (P36941, Invitrogen).

The COS7 cells were cultured in DMEM (GIBCO) containing 10% (V/V) fetal bovine serum (GIBCO) and 100 U ml$^{-1}$ penicillin and 100 µg ml$^{-1}$ streptomycin solution (PS, GIBCO) at 37 °C in an incubator with 95% humidity atmosphere and 5% $CO_2$. The selected cell lines were seeded on no. 1.5H cover glasses (CG15XH, Thorlabs) before they were grown to a suitable density (24 h) via incubation in a humid atmosphere containing 5% (V) $CO_2$ at 37 °C. The cells were stained in DMEM containing 500 nM MitoTracker Orange (M7510, ThermoFisher) and 0.5% dimethyl sulfoxide for 20 min in a $CO_2$ incubator. The cells were washed with PBS and fixed in 4% formaldehyde (R37814, Invitrogen) for 15 min at room temperature. The coverslip was sealed onto the cavity of the slide. The coverslip was sealed on the slide with the prolong antifade mountant (P36941, Invitrogen). For live COS7 cell imaging, the COS7 cells were seeded on µ-Slide 8 Wellhigh (80806, ibidi). The COS7 cells were stained in DMEM containing 500 nM IMMBright660 and 0.5% dimethyl sulfoxide for 20 min in a $CO_2$ incubator. Then, the cells were kept in DMEM for live-cell imaging without washing. For multicolor COS7 cells imaging, COS7 cells were cultured, fixed with a glutaraldehyde-based extraction-fixation procedure, and stained for actin (phalloidin-Atto 488, Sigma 49409), clathrin heavy chain (abcam ab21679; donkey anti-rabbit Alexa Fluor 555, ThermoFisher Scientific A31572) and alpha-tubulin (Sigma T5168 and T6199; donkey anti-mouse Alexa Fluor 647, ThermoFisher Scientific A31571) according to a previously published protocol[22].

The mouse kidney sections used to reconstruct the actin filaments were purchased from ThermoFisher. Actin filaments were labeled by Alexa Fluor 568 using Prolong Diamond to embed.

The COS7 cells used to reconstruct the nuclear pore complex were purchased from GATTA Quant. Anti-Nup was labeled by Alexa Fluor 555 using Prolong Diamond to embed.

The bovine large artery endothelial cells used to reconstruct the multicolor sample were purchased from ThermoFisher. Mitochondria were labeled with red-fluorescent MitoTracker Red CMXRos, F-actin was stained using green-fluorescent Alexa Fluor 488 phalloidin and blue-fluorescent 4,6-diamidino-2-phenylindole was used to label the nuclei using Prolong Diamond to embed.

### Data acquisition

We obtain data based on the commercial OMX-SIM system (DeltaVision OMX SR, GE) using an oil immersion objective (Olympus, ×60 1.4 numerical aperture (NA)) and a commercial N-SIM system (Nikon) using an oil immersion objective (CFI Apochromat, ×100 1.49 NA).

For the OMX system, 3DSIM sequences were performed with a pixel size of 80 and 125 nm in the *xoy* and *xoz* plane (five phases, three angles and 15 raw images per plane). And for the N-SIM system, 3DSIM sequences were performed with a pixel size of 65 and 120 nm in the *xoy* and *xoz* plane (five phases, three angles and 15 raw images per plane).

Samples in Fig. 2c and Extended Data Fig. 8a were obtained from open-source data in SIMnoise (v.1.0)[15,23] (https://data.4tu.nl/articles/_/12942932). COS7 cells in Fig. 2e were obtained from C. Leterrier (Aix Marseille University) using the Nikon N-SIM system equipped with a 100X, NA 1.49 objective and a Hamamatsu Fusion BT camera. Samples in Extended Data Figs. 2a, 3a and 4b were obtained from open-source data in fairSIM (v.1.5.0)[5] (http://www.fairsim.org/). The simulated structure in Extended Data Fig. 5a was obtained from open-source 3D structure data (v.1.2)[24] (https://github.com/Biomedical-Imaging-Group/GlobalBioIm). The simulated resolution test image in Extended Data Fig. 5b is ISO12233:2000 (Imatest). Extended Data Fig. 6a was obtained from AO-3DSIM (v.1.0.0)[14] (https://www.ebi.ac.uk/biostudies/studies/S-BSST629). Extended Data Fig. 6b was obtained from 4BSIM (v.1.0)[16] (https://zenodo.org/record/6727773).

### Raw image format

Open-3DSIM supports three data formats including OMX (*.dv/*.tif/*.tiff), N-SIM (*.nd2/*.tif/*.tiff) and some home-built systems (*.tif/*.tiff). The picture sequence for OMX is phase, depth, channel, time, then angle. The picture sequence for N-SIM is image (angle in rows and phase in columns), depth, channel, then time. The picture sequence for home-built systems is phase, angle, depth, channel, then time. We suggest using samples of six or more layers for reconstruction to obtain optimal results.

### Image processing

The frequency domain and SNR of images were generated by ImageJ. HiFi-SIM (v.1.01)[7], SIMnoise (v.1.0)[15], OMX, AO-SIM (v.1.0.0)[14] and 4BSIM (v.1.0)[16] are used for comparison. The reconstructions of Open-3DSIM were processed with MATLAB (v.2018a, 2021a and 2021b) and ImageJ. Figures were plotted with Imaris (v.9.0.1), Visio (v.2016), Origin (v.2021b) and Adobe Illustrator (v.2020).

### Calibration of the illumination nonuniformity

Fluorescent beads (ThermoFisher Scientific) of 200 nm in diameter were prepared in a fixed slide[20]. The beads were imaged by OMX for three angles with five phases in each angle. By averaging images of the five phases, wide-field images of each angle can be obtained. To correct the light intensity at each point, the beads should be thick on the focal plane. Then interpolation is used to compensate the gap of beads to form a light intensity map of $calib_{ang1}$, $calib_{ang2}$ and $calib_{ang3}$. The calibration ratio of calib1 and calib2 can be expressed as:

$$calib_1 = calib_{ang2}/calib_{ang1}, calib_2 = calib_{ang3}/calib_{ang1}$$

### Calibration of MAE and SNR

MAE and SNR in Fig. 2b are used to evaluate the reconstruction of Open-3DSIM, SIMnoise and OMX-SIM. After image calibration to compensate for vibrating during shooting, the corresponding results of high SNR under different algorithms are used as ground truth $y(i)$, so the images of low SNR $y'(i)$ are calculated using MAE:

$$MAE = \frac{1}{m}\sum_{i=1}^{m}|y(i) - y'(i)|$$

where $i$ denotes the $i$th pixel of whole $m$ pixels, and SNR is used to evaluate the ability to remove the defocused background of different algorithms. We choose the part with signal and the part without signal to solve its mean value and variance, and use the following formula to solve SNR (dB):

$$SNR(dB) = 10\log_{10}\frac{mean(signal) - mean(noise)}{s.d.(noise)}$$

### Reporting summary

Further information on research design is available in the Nature Portfolio Reporting Summary linked to this article.

## Data availability

The supplementary data, parameters, corresponding comparisons and the install video of the ImageJ version have been uploaded on Figshare (https://figshare.com/articles/dataset/Open_3DSIM_DATA/21731315)[23,25].

## Code availability

Software, test data and detailed user guides for Open-3DSIM (MATLAB, ImageJ and Exe) have been uploaded to GitHub (https://github.com/Cao-ruijie/Open3DSIM). We claim an Apache license for Open-3DSIM.

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

## Acknowledgements

P.X. acknowledges the funding support from the National Key R&D Program of China (grant no. 2022YFC3401100) and the National Natural Science Foundation of China (grant nos. 62025501, 31971376 and 92150301). We thank the National Center for Protein Sciences at Peking University in Beijing, China, for assistance with GE DeltaVision OMX super-resolution imaging. We thank Antone A. Bajor (University of California Santa Cruz) for helpful discussion. C.L. would like to thank the Neuro-Cellular Imaging Service and Nikon Center for Neuro-NanoImaging at INP, funded by CPER-FEDER (PlateForme NeuroTimone PA0014842).

## Author contributions

R.C. and P.X. conceived the idea of Open-3DSIM. P.X. supervised the research. R.C., P.X., Y.L., X.C. and M.G. deduced the theory of Open-3DSIM. R.C. and M.L. designed and conducted most of the experiments. M.L., X.G., C.L., B.G., S.J. and Y.F. prepared and imaged samples. R.C., P.X., Y.L., X.C., Y.H., M.L., M.G. and X.X. discussed the theory of Open-3DSIM. R.C. and X.C. discussed the optimization of parameter estimating. R.C. and Y.L. conducted data spectrum optimization. R.C. wrote the code of Open-3DSIM. R.C., Y.L., M.L., Y.H. and P.X. wrote the paper with input from all authors.

## Competing interests

P.X. and R.C. are inventors on a filed patent application related to this work (ZL202211605447.2). The other authors declare no competing interests.

## Additional information

**Extended data** is available for this paper at https://doi.org/10.1038/s41592-023-01958-0.

**Correspondence and requests for materials** should be addressed to Peng Xi.

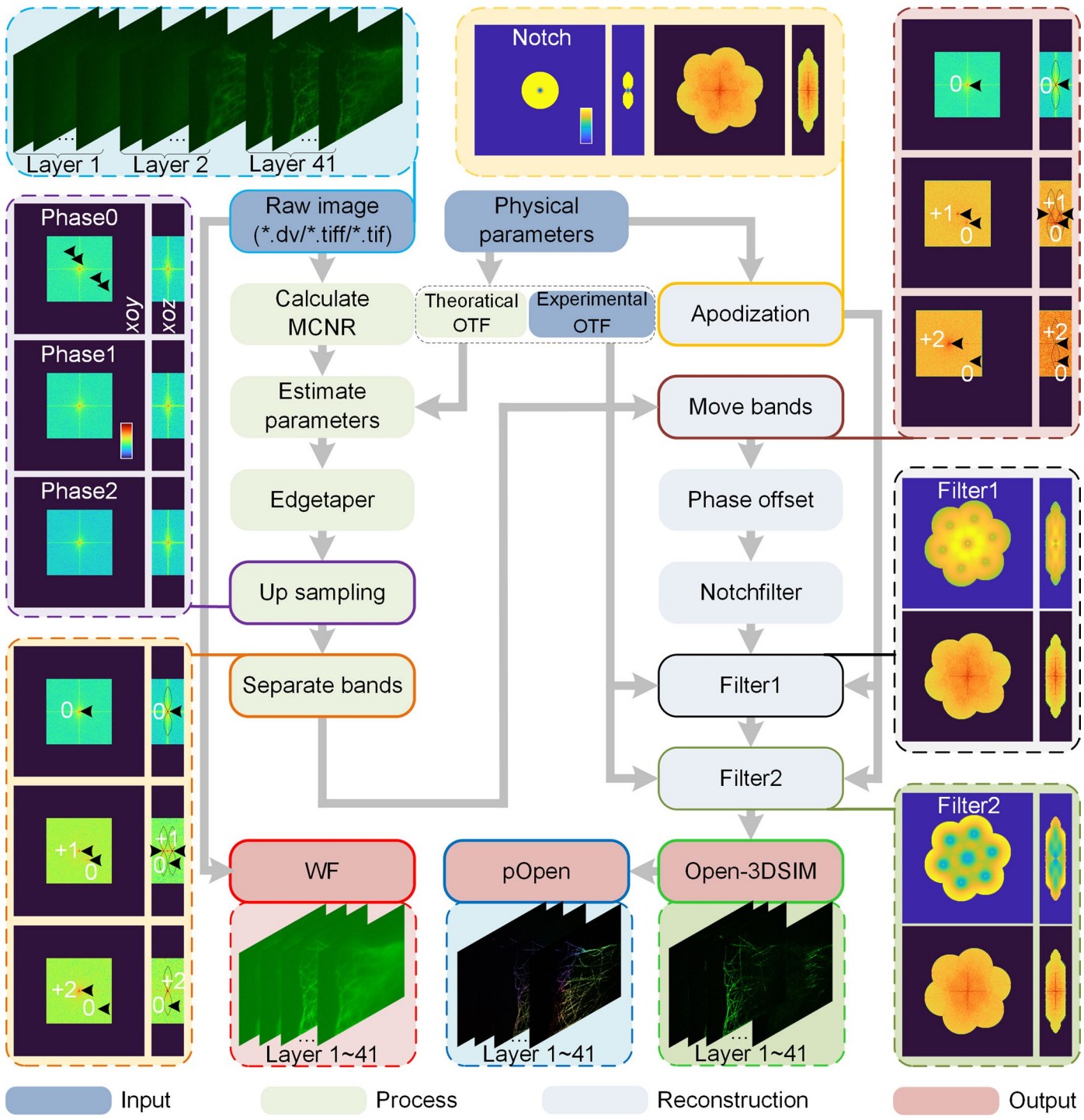

**Extended Data Fig. 1 | Algorithm flow and intermediate operations.** At color bars, in the format 'Intensity [a.u.]'.

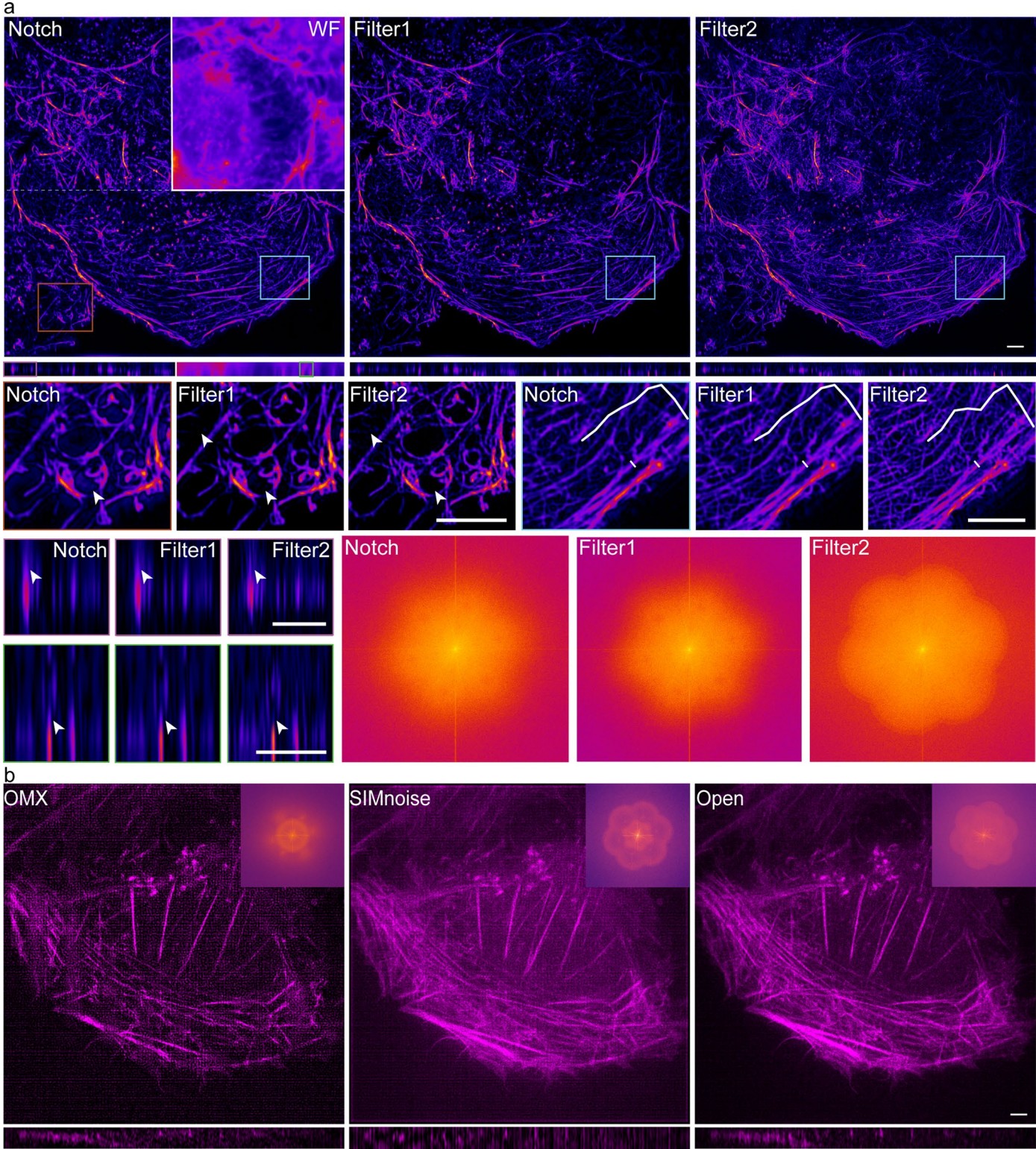

**Extended Data Fig. 2 | Effect of spectrum optimization and comparisons between different algorithms. a**, Effect of spectrum optimization including the reconstruction after notchfilter, after Filter1, and after Filter2, taking actin filament in liver sinusoidal endothelial cells (LSECs) from fairSIM as an example. The white arrows show that after two filters, the artifacts become fewer, and the white profile shows that high-frequency information is maintained. **b**, The reconstruction results and spectrums (embed top right) between different algorithms, taking actin filament in Fig. 2a as an example. Scale bar: 2 μm (top three lines and bottom two lines), 1μm (middle two lines). Scale on z-axis: **a**, 7 layers, **b**, 13 layers, 0.125 μm per layer.

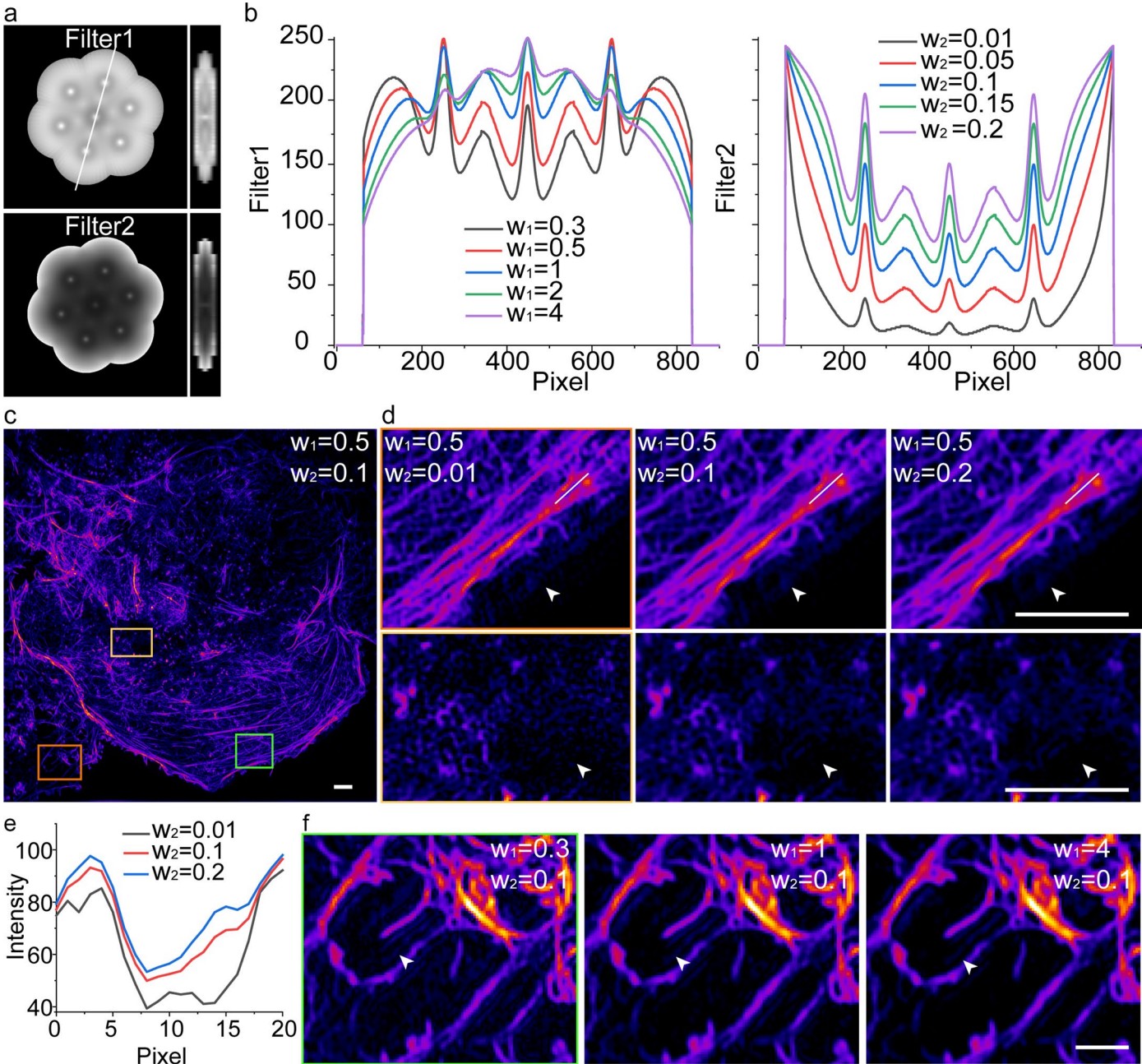

**Extended Data Fig. 3 | The effect of different parameters on reconstructed results. a**, The diagram of Filter1 and Filter2. **b**, The effect of different parameters ($w_1$ and $w_2$) on the profile of Filter1 and Filter2 (white line in a).

**c**, The reconstructed result of actin filament with $w_1 = 0.5$, $w_2 = 0.1$. **d**, The effect of different $w_2$. **e**, The profile in **d** with different $w_2$. **f**, The effect of $w1$. Scale bar: 2 μm. Scale on z-axis: 7 layers, 0.125 μm per layer.

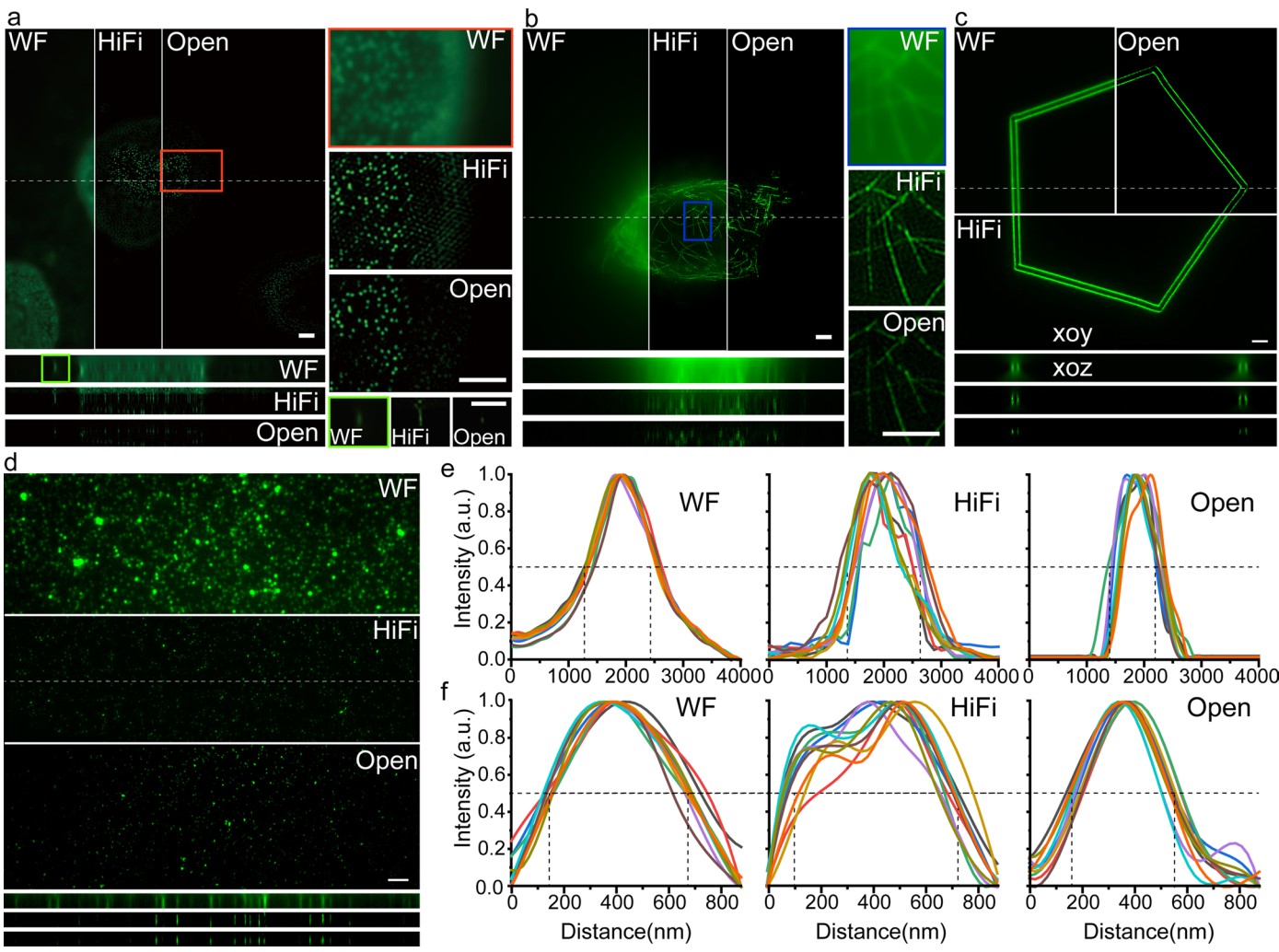

**Extended Data Fig. 4 | Comparison between single-layer and multi-layer SIM reconstruction. a**, Nuclear pore complex shot by OMX. **b**, Actin filament in U2OS osteosarcoma cells from fairSIM. Quantitative resolution calibration of WF, HiFi-SIM, and Open-3DSIM, including results of **c**, Argolight, and **d**, sparse fluorescent beads. And the fitted profile on z-axis of **e**, Argolight, and **f**, sparse fluorescent beads to quantify the half-height full-size and resolution based on 10 randomly selected points. Scale bar: 2 μm. Scale on z-axis: **a**, 17 layers, **b**, 8 layers, **c**, 33 layers, **d**, 8 layers, 0.125 μm per layer.

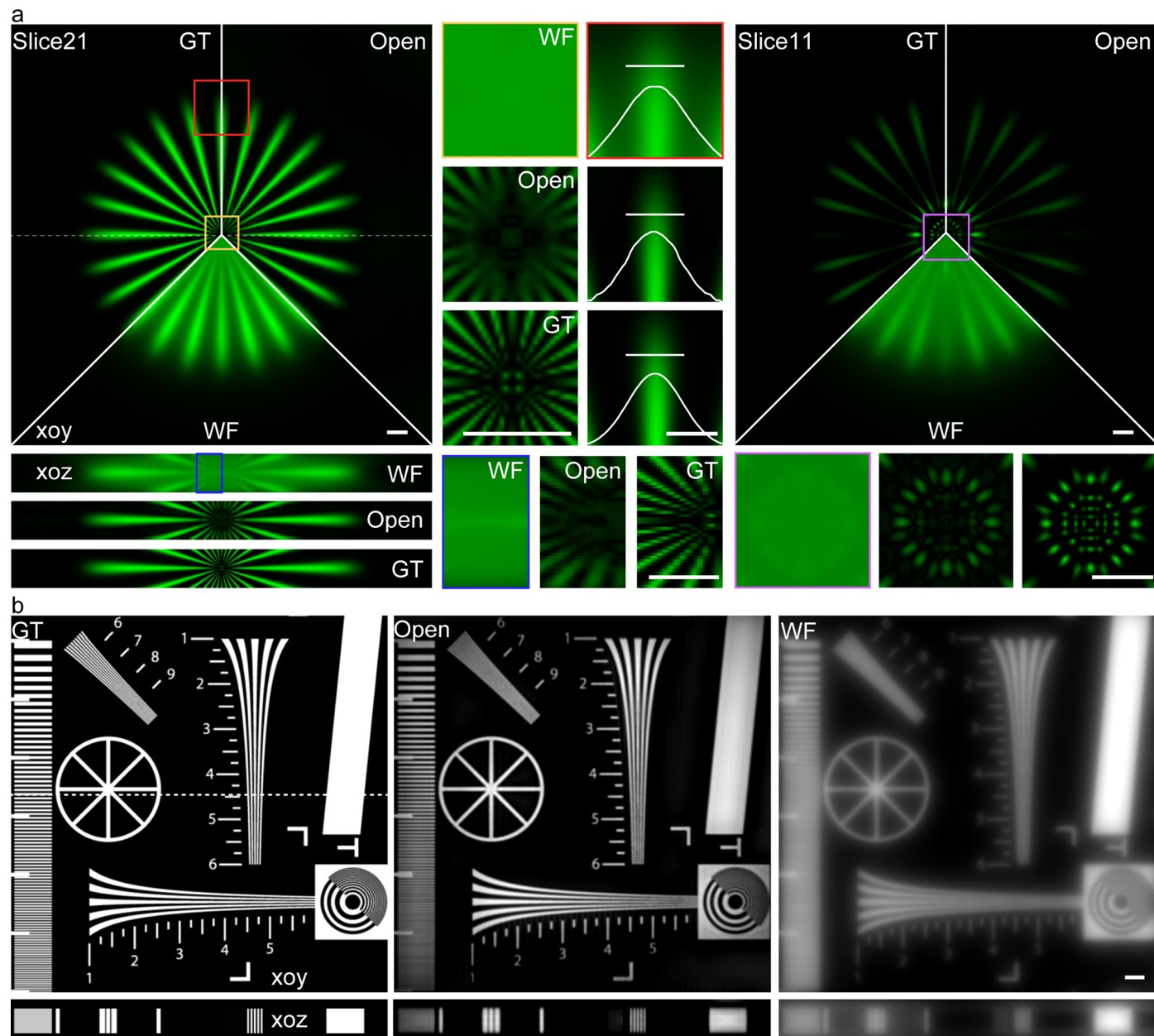

**Extended Data Fig. 5 | Simulation of Open-3DSIM. a**, The comparison of groud-truth image (GT), WF, and Open-3DSIM and the comparison between different $z$ slices of slice21 and slice11 using an open-source 3D structure. **b**, The comparison of GT, WF, and Open-3DSIM using a resolution test image. Scale bar: 2 µm. Scale on z-axis: **a**, 33 layers (0.125 µm per layer), **b**, 41 layers (0.125 µm per layer).

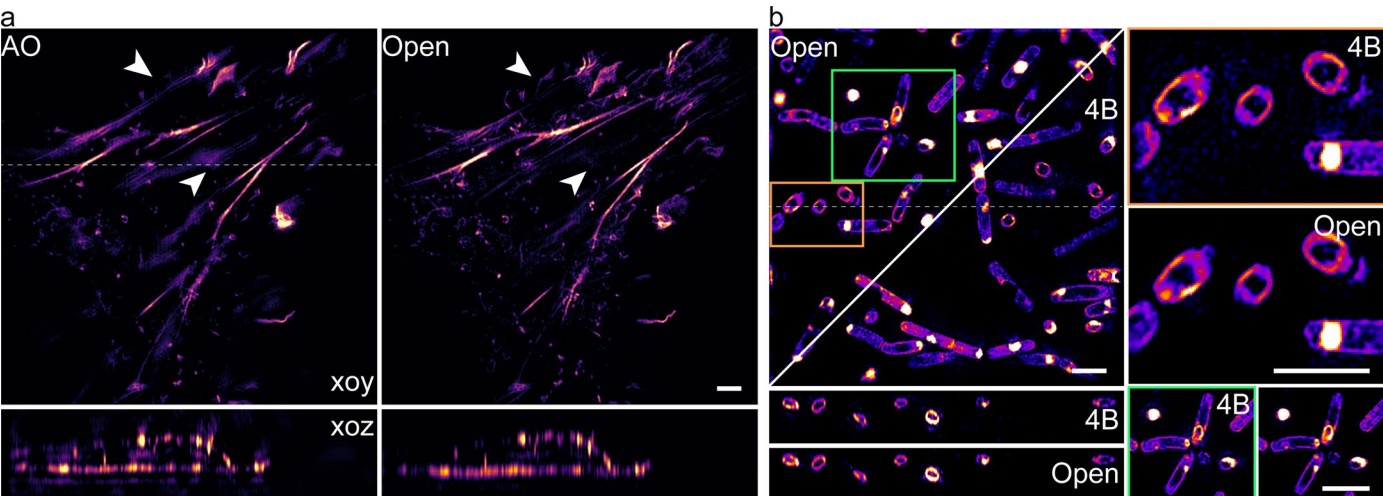

**Extended Data Fig. 6 | Comparison of traditional Weiner-based 3DSIM algorithms including AO-3DSIM and 4BSIM. a**, The actin of α-TN4 lens epithelial cell, whose reconstruction is derived from AO-3DSIM. **b**, The membrane of sporulating B. subtilis, whose reconstruction is derived from 4BSIM. Scale bar: 2 μm. Scale on z-axis: **a**, 26 layers, 0.2μm per layer, **b**, 32 layers, 0.125μm per layer.

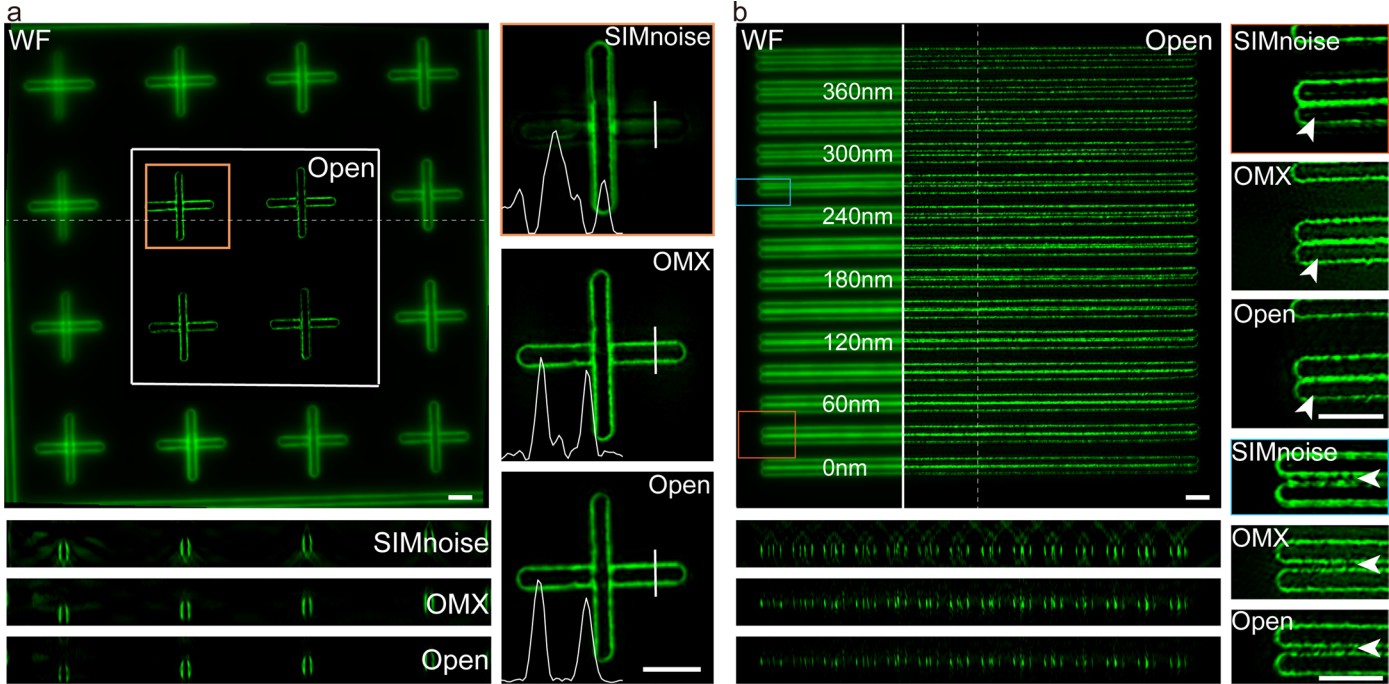

**Extended Data Fig. 7 | Comparison of various 3DSIM algorithms on Argo-SIM v1 test slide (Argolight, France).** including **a**, cross structure and **b**, linear structure (The interval of every line is from 0, 30, 60 to 390 nm). Scale bar: 2 μm. Scale on z-axis: **a**, 26 layers, **b**, 33 layers, 0.125 μm per layer.

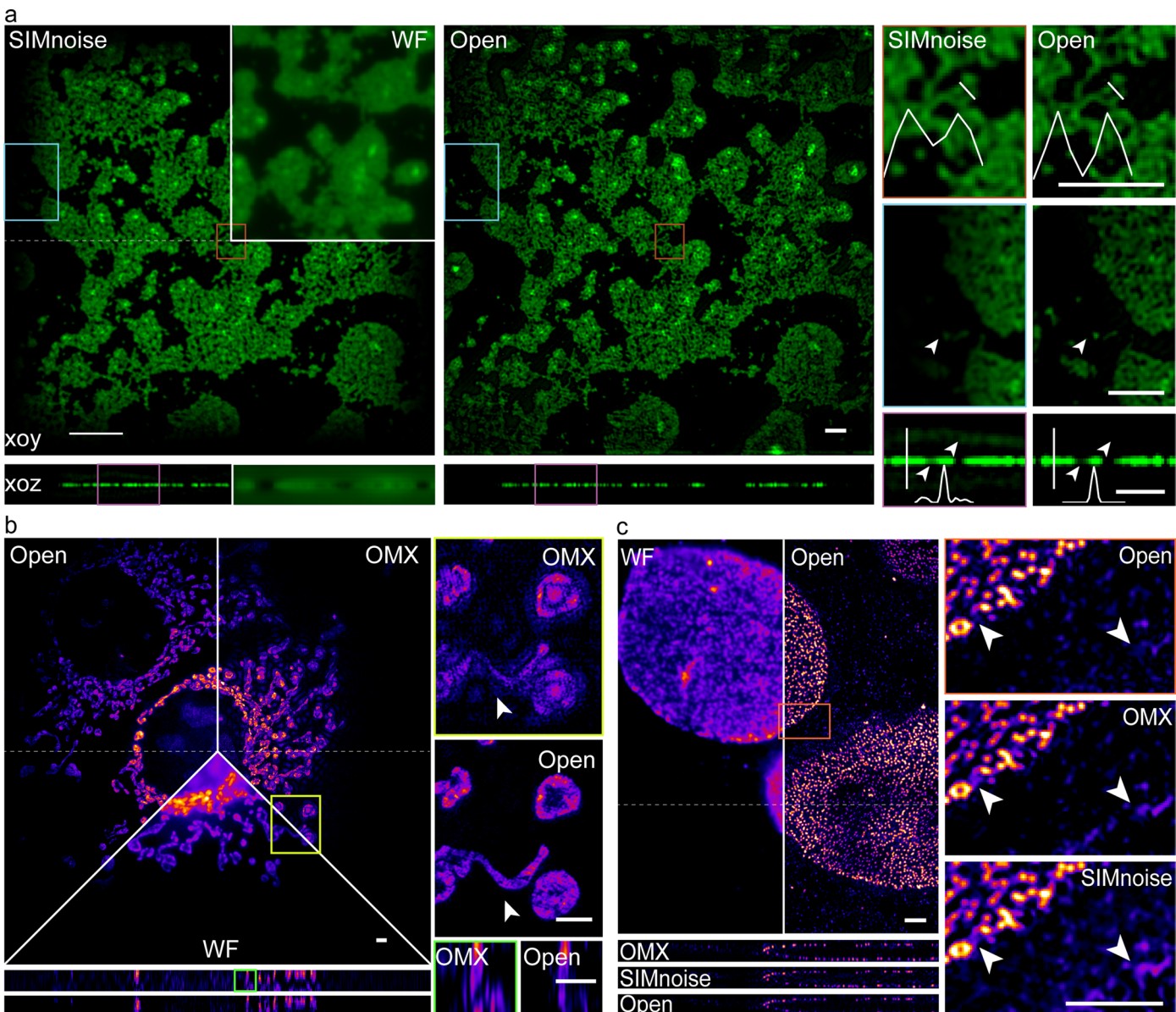

**Extended Data Fig. 8 | Comparison of various 3DSIM algorithms on various samples. a**, Beads with the state-of-art reconstruction of SIMnoise[1] under low SNR. **b**, Mitochondrial structure under low SNR. **c**, Nuclear pore complex under high SNR. Scale bar: 2 µm. Scale on z-axis: **a**, 33 layers, **b**, 9 layers, **c**, 25 layers, 0.125µm per layer.

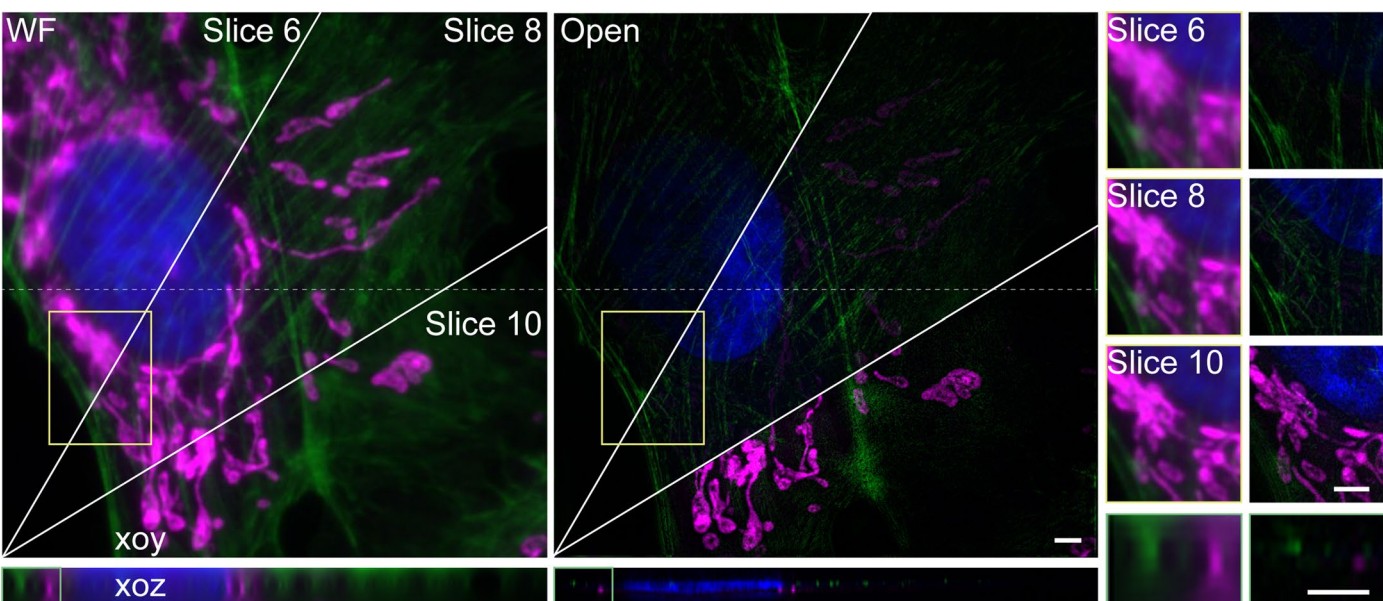

**Extended Data Fig. 9 | Reconstruction on a multi-color sample.** Reconstruction on a multi-color sample of the nucleus, actin filament, and mitochondrion, excited at 405, 488, and 568 nm in wavelength, respectively. And the comparison of WF and Open-3DSIM on different *z*-slice of Slice 6, 8, and 10. Scale bar: 2 μm. Scale on z-axis: 17 layers, 0.125 μm per layer.

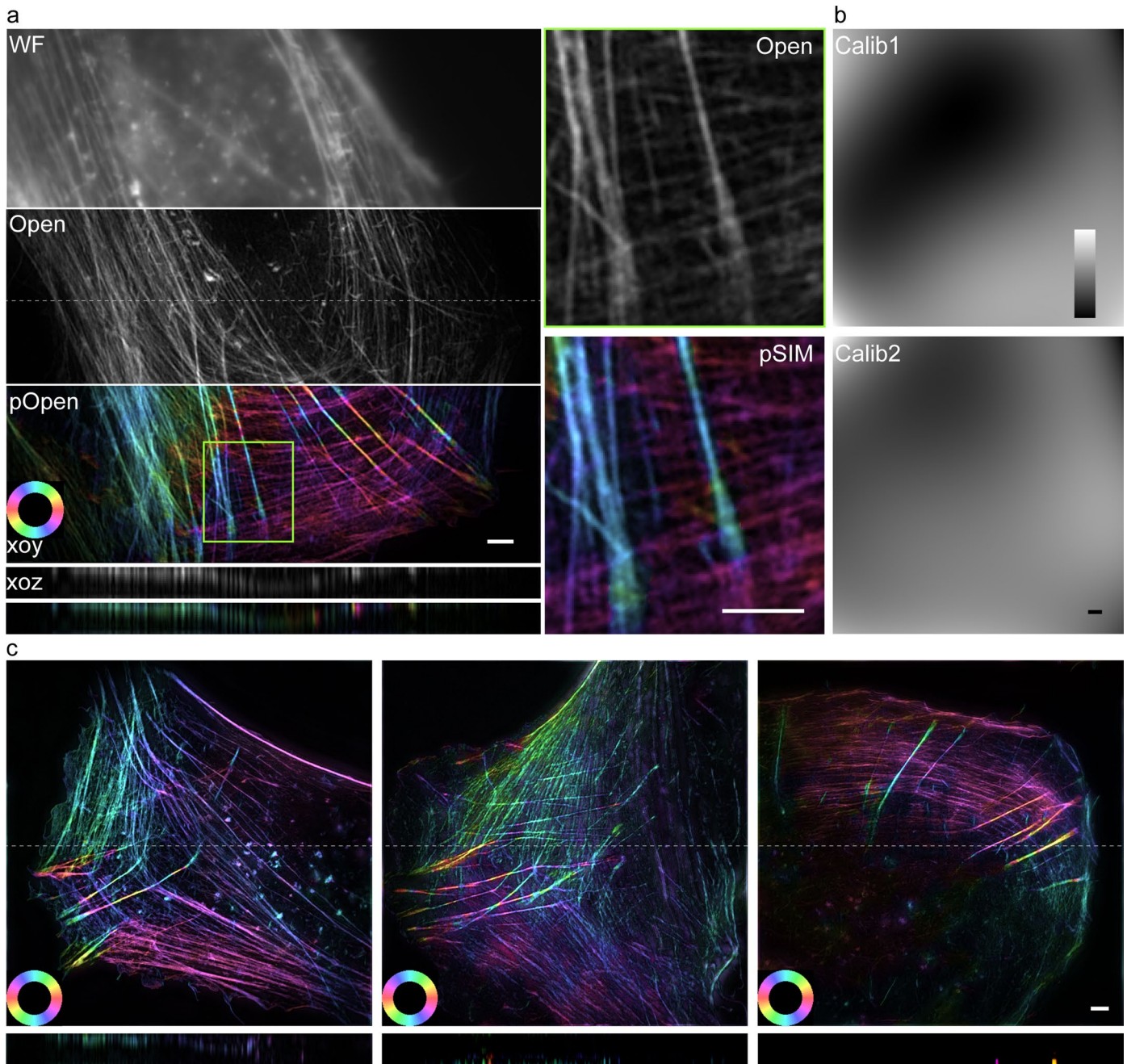

**Extended Data Fig. 10 | Reconstruction and dipole orientation imaging of actin structure. a**, Comparison between wide-field image, Open-3DSIM, and pOpen. **b**, Intensity calibration of different illumination angles. The intensity ratio between angle2 and angle1 is described as 'Calib1', and the intensity ratio between angle3 and angle1 is described as 'Calib2'. **c**, Reconstruction of more actin structure. Scale bar: 2 μm. Scale on z-axis: 9 layers, 0.125 μm per layer. At colour bars, in the format 'Angle [rad]'.

# Reporting Summary

## Statistics

For all statistical analyses, confirm that the following items are present in the figure legend, table legend, main text, or Methods section.

| n/a | Confirmed | |
|---|---|---|
| ☐ | ☒ | The exact sample size (*n*) for each experimental group/condition, given as a discrete number and unit of measurement |
| ☒ | ☐ | A statement on whether measurements were taken from distinct samples or whether the same sample was measured repeatedly |
| ☒ | ☐ | The statistical test(s) used AND whether they are one- or two-sided<br>*Only common tests should be described solely by name; describe more complex techniques in the Methods section.* |
| ☐ | ☒ | A description of all covariates tested |
| ☒ | ☐ | A description of any assumptions or corrections, such as tests of normality and adjustment for multiple comparisons |
| ☒ | ☐ | A full description of the statistical parameters including central tendency (e.g. means) or other basic estimates (e.g. regression coefficient) AND variation (e.g. standard deviation) or associated estimates of uncertainty (e.g. confidence intervals) |
| ☒ | ☐ | For null hypothesis testing, the test statistic (e.g. *F*, *t*, *r*) with confidence intervals, effect sizes, degrees of freedom and *P* value noted<br>*Give P values as exact values whenever suitable.* |
| ☒ | ☐ | For Bayesian analysis, information on the choice of priors and Markov chain Monte Carlo settings |
| ☒ | ☐ | For hierarchical and complex designs, identification of the appropriate level for tests and full reporting of outcomes |
| ☒ | ☐ | Estimates of effect sizes (e.g. Cohen's *d*, Pearson's *r*), indicating how they were calculated |

*Our web collection on statistics for biologists contains articles on many of the points above.*

## Software and code

Policy information about availability of computer code

Data collection
We obtain data based on the commercial OMX-SIM system (DeltaVision OMX SR, GE, USA) using an oil immersion objective (Olympus, Japan, ×60 1.4 NA) and commercial N-SIM system (Nikon, Japan) using an oil immersion objective (CFI Apochromat, Japan, ×100 1.49 NA).
For the OMX system, 3D-SIM sequences were performed with a pixel size of 80nm and 125nm in the xoy and xoz plane (5 phases, 3 angles, and 15 raw images per plane). And for the N-SIM system, 3D-SIM sequences were performed with the pixel size of 65nm and 120nm in the xoy and xoz plane (5 phases, 3 angles, and 15 raw images per plane).
Samples in Fig. 2(c), S. Extended Data Fig. 8(a) are obtained from open-source data in SIMnoise (v1.0): https://data.4tu.nl/articles/_/12942932. Cos7 cell in Fig. 2(e) is obtained from Dr. Christophe Leterrier (Aix Marseille University) using the Nikon system. Samples in Extended Data Fig. 2(a), Extended Data Fig. 3(a), and Extended Data Fig. 4(b) are obtained from open-source data in fairSIM(v1.5.0): http://www.fairsim.org/. The simulated structure in Extended Data Fig. 5(a) is obtained from open-source 3D structure data(v1.2) at https://github.com/Biomedical-Imaging-Group/GlobalBioIm. The simulated resolution test image in Extended Data Fig. 5(b) is ISO12233:2000 (Imatest, American). Extended Data Fig. 6(a) is obtained from AO-3DSIM (v1.0.0)(https://www.ebi.ac.uk/biostudies/studies/S-BSST629). Extended Data Fig. 6(b) is obtained from 4BSIM (v1.0)16 (https://zenodo.org/record/6727773).

Data analysis
We use SIMnoise(v1.0), OMX-system, HiFi-SIM(v1.01), AO-3DSIM(v1.0.0) and 4BSIM(v1.0) for comparison. Image decorrelation(v1.1.8), PSFj(July 28, 2014), and Fiji is used to analyse the reconstruction results. The results of the statistics are generated by Imaris(v9.0.1), Visio(2016), Origin (2021b), and Adobe Illustrator(2020).

For manuscripts utilizing custom algorithms or software that are central to the research but not yet described in published literature, software must be made available to editors and reviewers. We strongly encourage code deposition in a community repository (e.g. GitHub). See the Nature Portfolio guidelines for submitting code & software for further information.

## Data

Policy information about availability of data

All manuscripts must include a data availability statement. This statement should provide the following information, where applicable:

- Accession codes, unique identifiers, or web links for publicly available datasets
- A description of any restrictions on data availability
- For clinical datasets or third party data, please ensure that the statement adheres to our policy

> The supplementary data, parameters, corresponding comparisons, and the install video of the ImageJ version have been uploaded on Figshare (https://figshare.com/articles/dataset/Open_3DSIM_DATA/21731315).

## Human research participants

Policy information about studies involving human research participants and Sex and Gender in Research.

| | |
|---|---|
| Reporting on sex and gender | This information has not been collected. |
| Population characteristics | N/A |
| Recruitment | N/A |
| Ethics oversight | N/A |

Note that full information on the approval of the study protocol must also be provided in the manuscript.

# Field-specific reporting

Please select the one below that is the best fit for your research. If you are not sure, read the appropriate sections before making your selection.

☒ Life sciences          ☐ Behavioural & social sciences          ☐ Ecological, evolutionary & environmental sciences

For a reference copy of the document with all sections, see nature.com/documents/nr-reporting-summary-flat.pdf

# Life sciences study design

All studies must disclose on these points even when the disclosure is negative.

| | |
|---|---|
| Sample size | In this work, we present a 3D-SIM reconstruction tool. No sample-size based statistics is involved. |
| Data exclusions | No data were excluded. |
| Replication | We have upload the software, data and parameters. Using those, users can obtain the only result at any computer. So, we confirm the reproducibility of the experimental findings. |
| Randomization | This is not relevant to my study. Because we our data is not large, so the data is not allocated into groups. |
| Blinding | This is not relevant to my study. Because our duplicated reconstruction results are similar. |

# Reporting for specific materials, systems and methods

We require information from authors about some types of materials, experimental systems and methods used in many studies. Here, indicate whether each material, system or method listed is relevant to your study. If you are not sure if a list item applies to your research, read the appropriate section before selecting a response.

## Materials & experimental systems

| n/a | Involved in the study |
|---|---|
| ☐ | ☒ Antibodies |
| ☐ | ☒ Eukaryotic cell lines |
| ☒ | ☐ Palaeontology and archaeology |
| ☒ | ☐ Animals and other organisms |
| ☒ | ☐ Clinical data |
| ☒ | ☐ Dual use research of concern |

## Methods

| n/a | Involved in the study |
|---|---|
| ☒ | ☐ ChIP-seq |
| ☒ | ☐ Flow cytometry |
| ☒ | ☐ MRI-based neuroimaging |

## Antibodies

| Antibodies used | Alexa Flour 488, Alexa Fluor 555 and Alexa Fluor 568 |
|---|---|
| Validation | Alexa Flour 488 Phalloidin (A12379, USA) was purchased from Invitrogen directly.<br>The mouse kidney sections with Alexa Fluor 568 labeled was purchased from ThermoFisher(F24630, American) directly.<br>The COS7 cells with Alexa Fluor 555 labeled were purchased from GATTA Quant(GATTA-Cells 1C) directly. |

## Eukaryotic cell lines

Policy information about cell lines and Sex and Gender in Research

| Cell line source(s) | Human osteosarcoma U2-OS cell line(HTB-96) were purchased from ATCC.<br>The BPAE cells(F36924) were purchased from ThermoFisher.<br>The COS7 cells(GATTA-Cells 1C) were purchased from GATTA Quant. |
|---|---|
| Authentication | We directly purchased the cell line from ATCC/ThermoFisher/GATTA Quant which has been authenticated. |
| Mycoplasma contamination | We confirm that the cell line we used was tested negative for mycoplasma contamination. |
| Commonly misidentified lines<br>(See ICLAC register) | We don't use any commonly misidentified cell lines here. |

