## [Peer Review File · Nature Methods]

Peer Review Information

Manuscript Title: Open-3DSIM: an Open-source three-dimensional structured illumination microscopy reconstruction platform

Corresponding author name(s): Peng Xi

Editorial Notes: n/a

Reviewer Comments & Decisions:

Decision Letter, initial version:

Dear Peng,

Hello and happy new year!

Your Brief Communication, "Open-3DSIM: an Open-source three-dimensional structured illumination microscopy reconstruction platform", has now been seen by three reviewers. As you will see from their comments below, although the reviewers find your work of considerable potential interest, they have raised a number of concerns. We are interested in the possibility of publishing your paper in Nature Methods, but would like to consider your response to these concerns before we reach a final decision on publication.

We therefore invite you to revise your manuscript to address these concerns. As you will see, two referees mention unpublished open source software tools for 3D SIM reconstruction. While we do not require experimental comparisons to unpublished tools, these should be discussed appropriately in the main text and a textual comparison can be added (or perhaps a table comparing features).

We ask that you address the technical concerns, such as quantifying 3D resolution, and explain any caveats about comparing the 3D SIM reconstructions with 2D reconstructions.

We also ask that you better clarify the theoretical basis of your filtering steps and show the effects of parameterization on the reconstructions (refs 1 and 3).

We also ask that you update the software to improve usability, along the lines of referee 1.

We are not concerned with comments about "novelty", as we believe the tool to be of use to the community.

Given how many concerns were raised by the referees, we must state explicitly that we are not committed to sending the paper back to the reviewers until we've seen the revised version.

[Redacted]

This URL links to your confidential home page and associated information about manuscripts you may have submitted, or that you are reviewing for us. If you wish to forward this email to co-authors, please delete the link to your homepage.

We hope to receive your revised paper within three months. If you cannot send it within this time, please let us know. In this event, we will still be happy to reconsider your paper at a later date so long as nothing similar has been accepted for publication at Nature Methods or published elsewhere.

OPEN SCIENCE REQUIREMENTS

REPORTING SUMMARY AND EDITORIAL POLICY CHECKLISTS

Please note that these forms are dynamic ‘smart pdfs’ and must therefore be downloaded and completed in Adobe Reader. We will then flatten them for ease of use by the reviewers. If you would like to reference the guidance text as you complete the template, please access these flattened versions at <http://www.nature.com/authors/policies/availability.html>.

DATA AVAILABILITY

All novel DNA and RNA sequencing data, protein sequences, genetic polymorphisms, linked genotype and phenotype data, gene expression data, macromolecular structures, and proteomics data must be deposited in a publicly accessible database, and accession codes and associated hyperlinks must be provided in the “Data Availability” section.

To further increase transparency, we encourage you to provide, in tabular form, the data underlying the graphical representations used in your figures. This is in addition to our data-deposition policy for

specific types of experiments and large datasets. For readers, the source data will be made accessible directly from the figure legend. Spreadsheets can be submitted in .xls, .xlsx or .csv formats. Only one (1) file per figure is permitted: thus if there is a multi-paneled figure the source data for each panel should be clearly labeled in the csv/Excel file; alternately the data for a figure can be included in multiple, clearly labeled sheets in an Excel file. File sizes of up to 30 MB are permitted. When submitting source data files with your manuscript please select the Source Data file type and use the Title field in the File Description tab to indicate which figure the source data pertains to.

Please include a “Data availability” subsection in the Online Methods. This section should inform readers about the availability of the data used to support the conclusions of your study, including accession codes to public repositories, references to source data that may be published alongside the paper, unique identifiers such as URLs to data repository entries, or data set DOIs, and any other statement about data availability. At a minimum, you should include the following statement: “The data that support the findings of this study are available from the corresponding author upon request”, describing which data is available upon request and mentioning any restrictions on availability. If DOIs are provided, please include these in the Reference list (authors, title, publisher (repository name), identifier, year). For more guidance on how to write this section please see: <http://www.nature.com/authors/policies/data/data-availability-statements-data-citations.pdf>

CODE AVAILABILITY

Please include a “Code Availability” subsection in the Online Methods which details how your custom code is made available. Only in rare cases (where code is not central to the main conclusions of the paper) is the statement “available upon request” allowed (and reasons should be specified).

For more information on our code sharing policy and requirements, please see: <https://www.nature.com/nature-research/editorial-policies/reporting-standards#availability-of-computer-code>

MATERIALS AVAILABILITY

ORCID

Sincerely,
Rita

Rita Strack, Ph.D.
Senior Editor
Nature Methods

Reviewers' Comments:

Reviewer #1:
Remarks to the Author:
see attached PDF

[Attached]:

In their manuscript entitled "Open-3DSIM: an Open-source three-dimensional structured illumination microscopy reconstruction platform" authors Ruijie Cao et al. present both a variation and improvement of the original 3D SIM reconstruction algorithm (as introduced by Gustafsson et al. in 2008) as well as an open-source implementation of their improved algorithm.

Gustafsson's approach of 3D SIM reconstruction, based on frequency-domain unmixing, that allows for a 'direct' reconstruction (the equation system is fully defined and can be analytically solved) is well established and widely used in commercial SIM applications. Improving on this algorithm, in case of the authors by modifying and tuning the filtering steps, is thus a welcome improvement, especially useful for low SNR data in live cell imaging. The approach presented (supplementary note S1 & S2) seems sound and their results (fig. 2, especially the comparison with the reconstructions provided by the OMX) make it clear that these improvements seem to work.

The manuscript is well written, both the mathematical foundation of 3D SIM reconstruction (supplementary note S1) as well as the algorithms data flow (supplementary note S2) are easy to follow and very helpful to understand the algorithms. The main manuscript, both text and especially figures, are densely packed with information and could offer some more discussion and explanation. However, I assume this is due to the 'brief' format chosen that sets upper limits on text length and number of figures, so it is not easily changed.

Together with improvements of the algorithm itself, the authors provide an open-source implementation of this algorithm, based on MATLAB and connected also to a Fiji plugin for easier integration into existing workflows. I view this software package, its open source nature, general applicability to SIM data and also ease of use, as a core part of this publication.

As the authors state in the introduction, the SIM community is in need of an open-source, well established 3D reconstruction solution of 3-beam, 3D SIM datasets (as acquired on systems like the OMX, the Zeiss Elyra SIM or the Nikon N-SIM). I fully agree with this statement and am happy that they put effort into the work towards such a solution. In general, I fully support publication of the manuscript. However, in my opinion, some important points need to be addressed both in the manuscript and especially with the software and its packaging:

Manuscript:

- 1) In the introduction, the authors claim there is currently no general implementation of a 3D SIM reconstruction software widely available. To my knowledge, this is true in terms of software packages fully accompanied by an academic publication. However, at least the following packages are both freely available and regularly used by the SIM community, so the authors might want to mention them:
 - a) <https://github.com/scopetools/cudasirecon> A GPU-accelerated version of the original Gustafsson algorithms, as far as I know based on the same codebase.

Authored by Lin Shao and maintained (bugfixed, not necessarily feature improvements) by Talley Lambert. In active use by groups.

- b) https://github.com/Knerlab/SIM_Reconstruction A python-based implementation of 2D and 3D SIM reconstruction.

There is also a fairSIM version capable of 3D reconstruction (<https://github.com/fairsim/fairsim/tree/develop-3D-SIM>), but it is arguably still in a beta state (and development has become somewhat dormant), so I would not mention it here.

- 2) The authors mention the improved resolution provided by their algorithm, and visually this is supported by fig. 1 and 2. However, this claim might benefit from a more quantitative analysis, typically performed by applying Fourier ring correlation or image decorrelation analysis to the data, to get a quantitative resolution estimate. This should especially be performed for the comparison to the datasets reconstructed by the OMX-supplied software (SoftWORX, assumably).
- 3) The timelapse panel in fig. 2 seems to be cut off in my version of the manuscript, I do not know if this is intentional.

Software:

Unfortunately, I had some trouble getting the provided software to work on my machine. I was in the end able to run the software and confirm it generally reconstructs 3D SIM images. I however did not perform in-depth testing, as I am not sure my changes to the code are correct.

- 4) The provided Fiji plugin starts a GUI to control OpenSIM from within Fiji, but does not integrate with Fiji. In particular, input and output images are files on disks, not linked to the image stacks opened in Fiji. This makes interactive work and especially automation quite complex.
- 5) On my machine, the plugin crashed w/o a clear error indication. I did not investigate this further, but am happy to do so with a revised version of the manuscript and code. With the technical realization chosen by the authors, there are however two points that can be addressed to make crashes much less likely:
 - a) Checking for available memory, and providing clear error messages for out-of-memory problems. In my experience, 3D SIM reconstructions are demanding in terms of memory, so lower-end machines (8GB, even 16GB of RAM) will often run out of memory.
 - b) Since bundling with the MATLAB runtime is used, checking the code on a machine where no MATLAB is installed, and only the install guide is followed (easy to do in a virtual machine that can be reset). This avoids the problem of libraries missing on machines that do not have MATLAB installed.
- 6) The actual implementation of the algorithm is provided in MATLAB. Both versions (the one provided with the review material and the one on github) crashed with a missing reference to the 'phase' function in the fit_peak and find_illumination routines. Changing the 'phase' to the 'angle' function here (which seems to be the value to be retrieved

here) got me to a working version of the code, and reconstructions look reasonable. However, I would of course like to cross-check with the authors if this is the correct fix for this bug, and how it occurred.

- 7) Software repositories like *github* typically contain source code directly, so changes can be tracked and collaboration on code is possible. The repository provided by the authors, however, contains compressed archives (*rar* and *zip*), that then contain a copy of the code each. I would recommend changing this to a more standard format, where the repository contains the pure source code. To provide binary distributions (like *zip* files that are convenient for the end user), *github* offers the 'release' feature, where such data can be attached to (each version of) the source code.
- 8) There seems to be no clear license attached to the source code. As the authors aim to provide an open-source solution, I would recommend assigning one of the standard open-source software licenses (*GPL*, *BSD-2*, ...) to their code. This should be stated both in the manuscript, in the source code repository and the distributed binary files.

Reviewer #2:

Remarks to the Author:

The paper by Ruijie Cao et al. open-sourced a 3DSIM super-resolution image reconstruction algorithm, named Open-3DSIM. Although there are several commercial 3D-SIM instruments, such as GE OMX, Nikon N-SIM, Zeiss Elyra-SIM, their image reconstruction remained a black box for SIM users and researchers. Like the first open-source 2DSIM published in Nature Comm. in 2016 which triggered many follow-up studies to optimize the performance of reconstruction algorithms, I expect Open-3DSIM will have similar impacts in the field and eventually boost the application of 3DSIM in cell biology. Open-3DSIM algorithm is based on the established principle of 3DSIM. The authors adapted spectral filtering to diminish the artifacts. One novelty is the extension of 3DSIM to polarization-SIM so that SIM in six dimensions ($xyz\theta\lambda t$) can be realized. The demonstrated reconstruction results outperform the commercial 3D SIM and previously published SIMnoise. The supplementary code is complete, well-understandable, and easy to follow.

There are some issues that the authors should be addressed to make the paper more rigorous :

- (1) Since Open-3DSIM is designed for multi-layer 3D SIM and the advantage of multi-layer 3D SIM over single-layer is resolution doubling in z-direction. So the authors should give a quantitative resolution calibration using fluorescent beads or 3D patterns in Argolight. The current paper only shows decreased defocused backgrounds and artifacts without quantification.
- (2) Open-3D SIM used two-step frequency domain filtering to reduce the artifacts and improve the resolution. The filtering includes several designed functions and parameters. It is expected that different parameter settings have clear effects on the final results. So the authors may a table to list the parameters used in the demonstrated results and recommend how to set the parameters.
- (3) The scale bar for z-direction is not shown for all x-z section images. Is that the same for x-y direction? Generally, it is different.
- (4) There are some errors in the paper. For example, in line 122 of the main text, it should be "Supplementary note 4" instead of "Supplementary note 3".
- (5) In supplementary data, only one set of 3D image data is included. It is recommended that the authors also include other datasets, for example, data for Argolight and data for polarization 3DSIM.
- (6) In the Algorithm flow (Supplementary Note2), it shows three images of phase 0, phase 1 and phase 2. To remove the confusion, the authors may draw this figure with 5 images of phase 0-4 and also include the separate bands of -1,-2 order.

Reviewer #3:

Remarks to the Author:

This manuscript describes an open-source software package that can process 3D-SIM raw data to produce super-resolved images with the additional dipole-orientation info. While the package could in time prove to be a very useful toolkit for 3D-SIM community and the effort is much applaudable, I don't recommend its publication as a research article or brief in Nature Methods based on the following reasoning:

1. Lack of novelty.

1.a There is no significant breakthrough in the 3D-SIM reconstruction algorithm. What the authors proposed, as far as I can understand from the suppl notes, are not that different from the original Gustafsson et al 2008 paper. In fact, there might be some mistakes in SN1 about those filters and more on those below.

1.b The polarization or dipole-orientation analysis included in the package is completely based on a previously published technique.

1.c The authors try to sell their package as the only 3D-SIM open source package available. This is not true; there is the "cudasirecon" package downloadable from Github:

<https://github.com/scopetools/cudasirecon>

1.d The comparison between 3D-SIM processing results of this package with slice-by-slice 2D-SIM processing results from other published packages mostly misses the point: 3D-SIM data of course should be processed as a whole in 3D, as established in the original 3D-SIM paper, and of course it should outperform slice-by-slice 2D-SIM processing results. Although the latter's application in 3D-SIM can be attempted, its main value lies in such applications as single-layer 2D-SIM imaging without the benefit of TIRF configuration or very thin-layered samples. And it's not meant to compete with 3D-SIM in 3D performances.

2. Doubt-sowing mistakes

2.a Line 17-18 Page 3: the authors wrote "the modulation depth will rapidly decrease with the bias of the focal plane". Even this sentence has little to do the focus of the manuscript, it is nevertheless incorrect (especially so in the related Fig SN1(a), lower half) and thus gives me excuses to doubt the merit of the manuscript. The modulation depth should remain ~constant with defocus. The authors might be referring to the gradual damping of the axial components of the SIM pattern, as seen in Fig 3(b) of the Gustafsson et al 2008 paper. That damping effect was from the spatial incoherence introduced by the multi-mode fiber in that particular SIM setup. Regardless of coherent or incoherent light source, the lateral modulation depth should be more or less constant with defocus. What's worse, Fig SN1(a) depicts an axially confined SIM pattern, which cannot be farther from reality.

2.b From what's written in SN1 about the series of filters, I don't see a theoretical basis that support the plausibility of the algorithm. It seems that both Filter1 and Filter2 contain the OTF in the denominator, and wouldn't that mean the after applying them in series, the raw image would have been inverse-filtered twice and therefore the result be wrong? In SN3, related the spectrum filtering, why is the comparison with "Raw" and not with results from other 3D-SIM processing methods?

Author Rebuttal to Initial comments

Response to Reviewers' Comments

We very much appreciate the critical reading of our manuscript and valuable suggestions from the both editor and reviewers. We have carefully reviewed the comments and have revised the manuscript accordingly. To address all the comments, we reorganized the manuscript in the type of research article and underlined significant changes to facilitate the review of the revised manuscript. The responses to the comments are listed one by one as follows (in blue):

To editor:

Your Brief Communication, "Open-3DSIM: an Open-source three-dimensional structured illumination microscopy reconstruction platform", has now been seen by three reviewers. As you will see from their comments below, although the reviewers find your work of considerable potential interest, they have raised a number of concerns. We are interested in the possibility of publishing your paper in Nature Methods, but would like to consider your response to these concerns before we reach a final decision on publication. We therefore invite you to revise your manuscript to address these concerns.

Response: Dear editor, thank you for your support and interest in our manuscript "Open-3DSIM: an Open-source three-dimensional structured illumination microscopy reconstruction platform".

After posted on BioAxiv, our work has received widespread attention, and hundreds of experts and scholars have read our article. Many of them have used Open-3DSIM to perform 3DSIM reconstruction on various systems with appreciation and suggestions.

We very much appreciate the critical reading of our manuscript and valuable suggestions from the editor. We have carefully reviewed the comments and have revised the manuscript accordingly.

Q1: As you will see, two referees mention unpublished open source software tools for 3D SIM reconstruction. While we do not require experimental comparisons to unpublished tools, these should be discussed appropriately in the main text and a textual comparison can be added (or perhaps a table comparing features).

Response: We are very grateful for this helpful suggestion. We have carefully investigated several open-source 3DSIM algorithms, properly corrected the description in the manuscript, listed **Table**

E1 for comparison, and done experimental comparisons. We have added the discussion in **Manuscript** Page 2: Line 11-16 (yellow) and added **Supplementary Note 7** for the experimental comparisons.

The table to compare open-source 3DSIM algorithms is shown in **Table E1**. It can be seen that we first developed GUI within Fiji and Exe with three normal input formats, making it easy for biologists to use and compatible with different hardware systems. Then, our solution doesn't need initial parameters (such as frequency, and angle of illumination pattern) and experimental OTF, making it fully compatible with commercial or home-built 3DSIM systems. Last, we use adaptive parameter estimation and spatial spectrum optimization to achieve the optimal fidelity reconstruction with reduced artifacts, controlled noise, and retained weak information.

Table E1. Comparison of open-source 3DSIM algorithms.

Algorithm	Platform	Property	Purpose	Performance	Universality		
					Input formats	Initial parameters	OTF
Open-3DSIM	MATLAB Fiji (GUI) Exe (GUI)	Spectrum optimized and multi-platform	Optimize performance and Maximize user-friendliness	Artifact-reduced Noise-control	Three (OMX, NSIM, Home-built)	No	Experimental /Simulated
Cudasirecon	C++	Creation of wiener-based 3DSIM	Prove 3DSIM principle	Classical	One (OMX)	Need	Experimental
AO-3DSIM	Python	Traditional wiener-based 3DSIM	Adaptive optics to reduce artifacts	No optimization on algorithms	One (Home-built)	Need	Experimental /Simulated
SIMnoise	MATLAB	Iterative optimization denoising	Reconstruct well under low SNR	Noise-control	Two (OMX, Zeiss)	Need	Experimental /Simulated
3DSIM (4BSIM)	MATLAB	Traditional wiener-based 3DSIM	To compare 3DSIM with 4BSIM	No optimization on algorithms	One (Home-built)	Need	Experimental

Figure E1 | Comparison of traditional Wiener-based 3DSIM algorithms including AO-3DSIM and 4BSIM. a, the actin of α -TN4 lens epithelial cell, whose reconstruction is derived from AO-3DSIM. **b**, the membrane of sporulating *B. subtilis*, whose reconstruction is derived from 4BSIM. Scale bar: 2 μ m. Scale on z-axis: **a**, 26 layers, 0.2 μ m per layer, **b**,

32 layers, 0.125 μ m per layer.

According to **Table E1**, except that SIMnoise and Open-3DSIM did optimization on Weiner-based 3DSIM, Cudasirecon, AO-3DSIM, and 3DSIM part of 4BSIM are completely based on traditional Weiner 3DSIM for certain hardware purposes. So, it is the reason that our manuscript mainly compares Open-3DSIM with SIMnoise and commercial OMX systems. And in response to the suggestions, we add **Supplementary Note 7 (Fig. E1 here)** to do comparisons between Open-3DSIM, AO-3DSIM, and the 3DSIM part of 4BSIM (simplified as 4BSIM) compared with their well-adjusted reconstructed results. It can be shown that Open-3DSIM outperforms traditional Weiner-based 3DSIM algorithms including AO-3DSIM and 4BSIM with fewer artifacts and more retained weak information. And in **Fig. 2(a-d, g)** and **Supplementary 8-9**, we demonstrate its superiority compared with SIMnoise and OMX.

Figure E2 | The graphic user interface and operation interface of Open-3DSIM, including a, Fiji version, b, Exe application and c, MATLAB code.

And, we have added the GUIs for Open-3DSIM (**Fig. E2**) and other algorithms (**Fig. E3**) for your reference. It can be seen that Open-3DSIM fully considered the convenience for biologists

and hardware specialists. Open-3DSIM not only integrates the Fiji plugin to provide an additional and kind tool for biologists, but also provides the middle results of reconstruction for hardware specialists to evaluate their data.

What's more, Open-3DSIM is multi-platform, not limited by specific microscope workstations, and can be customized and adjusted according to user needs. And it is also extensible, it is fully compatible with other algorithm optimization methods based on regular terms or machine learning. Its superiority, universality, compatibility, and user-friendliness can provide a solid software guarantee for the future development of 3DSIM.

Action:

On Manuscript, Page 2, Line 9-22, revised the description;

On Manuscript, Page 3, Line 13-14, revised the description;

On Manuscript, Page 4, Line 15-18, revised the description;

On Manuscript, Page 8, Line 1-3, revised the description;

On Supplementary Note 7, SI Page 18, Line 1-16, added the comparison.

a

```

SireconDriver.m
16
17 clear all;
18
19 arg = struct(...
20 'files', 'E:\MATLAB_program\4BSIM\Sirecon\Data\Test',... % input file (on data folder in TIFF mode)
21 'outfile', 'out_3DSIM_IP35NA',... % output file (or filename pattern in TIFF mode)
22 'otffiles', 'E:\MATLAB_program\4BSIM\Sirecon\OTF\OTF_3DSIM_IP35NA_488.tiff',... % OTF file
23 'usecorr', [],... % flat-field correction file provided
24 'ndirs', 3,... % number of directions
25 'nphases', 5,... % number of phases per direction
26 'nordersout', 0,... % number of output orders; must be <= norders
27 'angle0', -0.19142,... % angle of the first direction in radians (1.27 NA: -1.3994; 1.35 NA: -0.19142)
28 'l0', [0.1999, 0.1994, 0.1994],... % line spacing of SIM pattern in microns
29 'na', 1.35,... % detection numerical aperture
30 'nlim', 1.406,... % refractive index of immersion medium
31 'nmed', 1.406,... % refractive index of sample medium
32 'iter_input', 1,... % iteration number for input RL deconvolution
33 'iter_output', 2,... % iteration number for output RL deconvolution
34 'zoomfact', 2,... % lateral zoom factor
35 'explofact', 1.0,... % artificially exploding the reciprocal-space distance between orders by this fi
36 'zoom', 1,... % axial zoom factor
37 'background', 100.0,... % camera readout background (be will be subtracted from input images)
38 'wiener', 0.001,... % Wiener constant
39 'linear_wiener', [],... % linear increment of Wiener constant for time-lapse imaging
40 'forcepsamp', [],... % orders forced to these values
41 'kangles', [2.94954, 1.90853, -2.28121],... % user given pattern vector k0 angles for all directions
42 'gamma0', 1.0,... % output apodization gamma; 1.0 corresponds to a triangular shape, lower numb
43 'savefiltered', [],... % save separated bands (in frequency domain) into a file and exit
44 'savealignedraw', [],... % save drift-fixed raw data (half Fourier space) into a file and exit
45 'saveoverlap', [],... % save overlap and overlap (real-space complex data) into a file and exit
46 'config', [],... % name of a file of a configuration.
47 'lambd0', 488,... % excitation wavelength: 488, 561
48 'wavelength', 525); % emission wavelength: 525, 607 (only used for TIFF files)
49

```

b

```

simnoise.m
24 % add directory with all subroutines to the Matlab path
25 addpath(genpath('..\helperfunctions'))
26 addpath(genpath('..\ofmatlab'))
27 addpath('..\lib\diplib\share\DIPIImage');
28 setenv('PATH', ['..\lib\diplib\bin', ';', getenv('PATH')]);
29 % Data reformat chapter
30
31 datparams.allSIMdatasets = {'20180211_6-layer_STD_512_T1_80ms_Fov3_46'}; % filename(s) of the structured illumination m
32 datparams.rootdir = './data/'; % input directory with raw data and output directory for preprocessed image data and pa
33 datparams.datadir = 'OMXdatafiles'; % input directory with sample data from OMX
34 datparams.otfdir = 'OMXdatafiles'; % input directory with the otf files
35
36 datparams.numangles = 3; % number of pattern angles, OMX system
37 datparams.numsteps = 5; % number of phase steps, OMX system
38 datparams.gain = 2.0; % default value gain found with single-shot gain estimation
39 datparams.offset = 50.0; % default value offset found with single-shot gain estimation
40 datparams.RMSD = 1.0; % default value rms readout noise
41
42 datparams.numchannels = 1; % number of color channels in dv-file, 4x88
43 datparams.selectwavelengths = 2; % select color channel, 1=red,2=green,3=blue,4=far-red
44 datparams.numframes = 1; % number of frames in time series
45
46 datparams.rawpixelsize = [80 80 125]; % pixel size and focal stack spacing (nm)
47 datparams.NA = 1.4; % objective lens NA
48 datparams.refmed = 1.47; % refractive index medium
49 datparams.refcov = 1.512; % refractive index cover slip
50 datparams.refim = 1.512; % refractive index immersion medium
51 datparams.wavelengthlib = [609 525 435 683]; % set of possible emission wavelengths for red/green/blue/far-red
52 datparams.wavelengthslib = [568 488 405 640]; % set of possible excitation wavelengths for red/green/blue/far-red
53 datparams.patternpitchlib = [435 426 397 488]; % set of approximate pattern pitch values OMX system for red/green/blue
54 datparams.allpatternangle_init = [45 105 -15] * pi / 180; % approximate values pattern angles for used OMX system
55 datparams.allpatternphases_init = 2 * pi * (1 - (0:(datparams.numsteps-1)) / datparams.numsteps); % initial value pattern ph
56

```

c

```

sim_3drecon_p36.py
21 def __init__(self, fnd, nph, nangles, wavelength, na):
22     self.img_stack = tf.imread(fnd) # order of images should be phases, angles, z-slices
23     # self.img_stack = self.subback(self.img_stack)
24     self.img_stack = np.pad(self.img_stack, ((2*nph*nangles, 2*nph*nangles), (0, 0), (0, 0)), 'const',
25     print('Image stack loaded successfully')
26     nz, nx, ny = self.img_stack.shape
27     self.nz = int(nz/nph/nangles)
28     self.nx = nx
29     self.ny = ny
30     self.mu = 1e-2 # Wiener parameter
31     self.wl = wavelength # in microns
32     self.cutoff = 1e-3 # remove noise below this relative value in freq. space
33     self.na = na # numerical aperture
34     self.dx = 0.089 # pixel size in microns
35     self.dz = 0.2 # z step in microns
36     self.nphases = nph
37     self.norders = 5
38     self.dpx = 1 / ((self.nx * 2) * (self.dx / 2)) # calculate pixel size in frequency space
39     self.dpz = 1 / ((self.nz * 2) * (self.dz / 2)) # calculate axial pixel size in frequency space
40     self.radius_xy = (2 * self.na / self.wl) / self.dpx # NA in pixels
41     self.radius_z = ((self.na * 2) / (2 * self.wl)) / self.dpz
42     self.strength = 1.
43     self.sigma = 8.
44     self.eta = 0.08
45     self.expn = 1.
46     self.axy = 0.8
47     self.az = 0.8
48     self.zoa = 10e-2
49     self.nangle = 0
50     self.psf = self.getpsf()
51     self.sepmat = self.sepmatrx()
52     self.meshgrid()
53     self.winf = self.window(self.eta)
54     self.apd = self.apod()
55     self.img_stack = self.img_stack.reshape(self.nz, nangles, nph, nx, ny).swapaxes(0, 1).swapaxes

```

d

```

reconpy
1 import os
2 from contextlib import contextmanager
3 from subprocess import run
4
5 import mrc
6 import numpy as np
7
8 # NOTE: this assumes that you have config files and otf files with the emission wavelength
9 # names in them... such as 'config528.txt' and 'otf528.otf'. You may place those files
10 # anywhere, but you should update the _DIR variables below
11
12 # path to your otf directory. Defaults to the same as the recon.py file
13 OTF_DIR = os.path.abspath(os.path.dirname(__file__))
14 # path to your config directory. Defaults to the same as the recon.py file
15 CONFIG_DIR = os.path.abspath(os.path.dirname(__file__))
16 # name of the reconstruction app
17 APP = "cudasirecon"
18
19
20 # you could of course do something more complicated with the config/otf finding
21 def get_otf_and_config(wave, otf_dir=OTF_DIR, config_dir=CONFIG_DIR):
22     otf = os.path.join(otf_dir, f"otf{wave}.otf")
23     config = os.path.join(config_dir, f"config{wave}.txt")
24     if not os.path.exists(otf):
25         print(f"Could not find otf file: {otf}")
26         return None
27     if not os.path.exists(config):
28         print(f"Could not find config file: {config}")
29         return None
30     return otf, config
31
32
33 def merge_files(file_list, outfile, delete=False):
34     data = [mrc.imread(fname) for fname in file_list]
35     waves = [0, 0, 0, 0]
36     for i, item in enumerate(data):

```

Figure E3 | The graphic user interface and operation interface of other 3DSIM algorithms, including a, 4BSIM, b, SIMnoise, c, AO-3DSIM, and d, Cudasirecon.

Q2: We ask that you address the technical concerns, such as quantifying 3D resolution, and explain any caveats about comparing the 3D SIM reconstructions with 2D reconstructions.

Response:

(1) quantifying 3D resolution

We have added **Supplementary Note 6** to quantitatively explain the improvement of the z-axis resolution of multi-layer 3D SIM compared with single-layer 3DSIM. As shown in **Fig. E4(a)** and **(b)**, taking the Argolight pentagon pattern and 100 nm fluorescent beads of 488nm excitation

wavelength as examples, we take 10 spatial points on Argolight and Beads to make the intensity-pixel curve (processing with spline interpolation) on the z -axis as shown in **Fig. E4(c)** and **(d)**.

We used the full-width at half-maximum (FWHM) analysis function of origin to obtain the half-height full-size of different algorithms of Arglight and Beads. It can be seen that the half-height and full-width on the z -axis of WF, single-layer 3D SIM, and multi-layer 3D SIM are 1218nm, 1412nm, and 765nm for Argolight, respectively. And the half-height and full-width on the z -axis of WF, single-layer 3D SIM, and multi-layer 3D SIM are 596nm, 525nm, and 387nm for Beads, respectively.

At the same time, we use the PSFj plugin to quantitatively analyze the resolution of beads, and get that the resolution of the xoy plane and z -axis of WF is 251nm, 717nm, the resolution of the xoy plane and z -axis of single-layer 3DSIM is 120nm, 708nm, and the resolution of the xoy plane and z -axis of multi-layer 3DSIM is 118nm, 344nm. Therefore, from a quantitative point of view, we show that 3DSIM achieves double 3D resolution compared with WF and double z -resolution compared with single-layer 3DSIM.

Figure E4 | Quantitative resolution calibration of WF, HiFi-SIM, and Open-3DSIM, including results of **a**, Argolight, and **b**, sparse fluorescent beads. Scale bar: $2\mu\text{m}$. Scale on z-axis: **a**, 8 layers, **b**, 33 layers, $0.125\mu\text{m}$ per layer.

(2) explain any caveats about comparing the 3D SIM reconstructions with 2D reconstructions

The multi-layer processing method should outperform the single-layer processing method, which is indeed not comparable. We agree that 3D-SIM data should be processed as a whole in 3D.

But as a counterexample, fairSIM and HiFiSIM do use a certain layer of multi-layer 3DSIM data for single-layer processing. And it is noteworthy that the role of single-layer 3DSIM in this paper is similar to that of wide-field. It is only used as a reference to show the improvement of z-axis resolution rather than emphasizing Open-3DSIM's superior performance. And in response to the reviewer's suggestion, we modified the description of the comparison between single-layer 3DSIM and multi-layer 3DSIM at the end of **Supplementary Note 5**: We would like to stress that our comparison here to illustrate the advantages of multi-layer 3DSIM in improving z-axis resolution compared with single-layer 3DSIM. 2DSIM (or single-layer 3DSIM) has lower phototoxicity and faster imaging speed. On the contrary, multi-layer 3DSIM has the ability of whole-cell imaging with resolution improvement on z-axis. Users should reasonably select the required imaging method according to their needs.

Action:

On Supplementary Note 5, SI Page 14, Line 6-9, added reconstruction of Argolight and beads.

On Supplementary Note 5, SI Page 15, Line 4-21, added z-axis resolution qualification and explain any caveats about comparing the 3D SIM reconstructions with 2D reconstructions.

Q3: We also ask that you better clarify the theoretical basis of your filtering steps and show the effects of parameterization on the reconstructions (refs 1 and 3).

Response:

(1) theoretical basis of filtering steps

We have revised **Supplementary Note 1** to make the theory clearer. And state a simplified explanation here. In SIM, due to the abnormal spectrum distribution of the combination, various artifacts are usually caused. One typical problem is honeycomb artifacts caused by the high-frequency peak, which is caused by the abnormal distribution of $C_{SR,0}(k)$ with six frequency peaks of $\pm 2^{\text{nd}}$ components and six frequency peaks of $\pm 1^{\text{st}}$ components. So, the first optimization of Open-3DSIM is to use a designed notch-filter to suppress the high-frequency peaks. It is noteworthy that the ideal spectrum of 3DSIM is the same form of wide-field, being smooth and even in the 3D domain. We think the ideal 3D spectrum is smooth and petal-shaped to be approached, and the distribution of directly combined spectrum after the notch-filter is like the sum of $OTF \cdot notch$. So, the first filter is described as $Filter1(k) = \frac{Apo}{OTF_{notch+w_1^2}}$ to make the spectrum approach to the ideal one. As shown in **Fig SN3**, applying $Filter1(k)$ can greatly suppress the patchy features, high-frequency noise, so as to suppress sidelobe artifacts, but the resolution is decreased because of the suppress of the edge of the petal-shaped spectrum. As a result, some weak information may be disappeared. What's more, with the involvement of w_1 , the approach to the ideal spectrum and the compensation for the previous notch-filter is not complete. So, we adapt an extra filter called $Filter2(k) = \frac{Apo}{OTF_{notch+w_2^2}}$ to continuously approach the ideal spectrum and increase the resolution of the reconstructed image, and the proportion of high-frequency component can be properly increased. $Filter2(k)$ will greatly maintain the weak

information. The cooperation of two filters can fill the spectrum hole caused by the previous notch-filter and enhance the smoothness of the 3D spectrum.

(2) the effects of parameterization on the reconstructions

We have added **Supplementary Note 4** to guide users to choose the parameters and list a table of **Table E2(Table SN1)** to list the parameters in all our reconstructions. In general, parameters in Open-3DSIM are usually fixed and easy to be adjusted, which brings convenience for users.

In our work, we set all reconstructions with *edegetaper* (edge smoothing) and attenuation (*att* in eq(9)) equal to 0. To preserve edge and high-frequency information, we also recommend users keep the default settings. However, for the original image with lots of edge information, a reasonable set of some *Edegetaper* (typically 1-10 pixels for 512×512 images) can help to reduce the artifacts; In the case of serious high-frequency noise, a reasonable set of Attenuation between 0 and 0.5 will help to reduce the noise. *Notchwidth1* (w to construct OTF_{notch}) and *Notchwidth2* (w to construct notch-filter) are notch range parameters related to image size, and *Notchdepth1* (d to construct OTF_{notch}) and *Notchdepth2* (d to construct notch-filter) are constant notch depth. All reconstructions in this article maintain the default parameters of these four variables, and users are also recommended to keep the default parameters. However, in the process of spectrum analysis after reconstruction, if honeycomb artifacts caused by an insufficient notch (obvious high-frequency modulation point) are found, *Notchdepth2* and *Notchwidth2* can be appropriately increased, and *Notchdepth1* and *Notchwidth1* can be appropriately reduced.

As to the users, we set w_1 and w_2 to help users to adjust parameters. As shown in **Figure E5(a)**, *Filter1* generally presents a low-pass filter while *Filter2* presents a high-pass filter. Larger w_1 will make *Filter1* more obvious in suppressing the high-frequency information in the OTF of each spectrum, and smaller w_2 will enhance the high-frequency part of the petal-shaped spectrum (**Figure E5(b)**).

Table E2 The parameters and data origin of all of the reconstructions in our work.

Figure	Exwavelength(nm)	w_1	w_2	Data origin
Fig. 1(c)	488	0.5	0.1	OMX
Fig. 2(a)	568	0.5	0.1	OMX
Fig. 2(c)	488	0.5	0.1	SIMnoise
Fig. 2(d)	488	0.5	0.1	OMX
Fig. 2(e)	488, 561, 609	0.5,0.5,0.8	0.1,0.1,0.1	N-SIM
Fig. 2(f)	647	0.5	0.1	OMX
Fig. 2(g)	568	0.5	0.1	OMX
Fig. SN3(a, b)	488	0.5	0.1	fairSIM, OMX
Fig. SN4	488	0.8	0.1	fairSIM
Fig. SN5(a, b)	488,488	0.5,0.5	0.1,0.1	OMX, fairSIM
Fig. SN5(c, d)	488,488	0.5,0.5	0.1,0.1	OMX, OMX
Fig. SN6(a, b)	488	0.5	0.1	Simulation

Fig. SN7(a, b)	647,488	0.5,0.5	0.1,0.1	AO-3DSIM, 4BSIM
Fig. SN8(a, b)	488,488	0.5,0.5	0.1,0.1	OMX
Fig. SN9(a, b, c)	488,568,568	0.5,0.5,0.5	0.02,0.1,0.1	SIMnoise, OMX, OMX
Fig. SN10	405,488,568	0.5,0.5,0.5	0.1,0.1,0.1	OMX
Fig. SN11	488	0.5	0.1	OMX

Filter1 is mainly used for OTF editing after abnormal reorganization in cooperation with the previous notch filter (**Figure E5(c)**). When w_1 is larger, the suppression of high-frequency noise at various spectral levels is more obvious, and the ability to suppress sidelobe artifacts is enhanced. Users can adjust w_1 to achieve a balance between resolution and artifact suppression (**Figure E5(f)**). Filter2 is mainly used to preserve weak signals and improve resolution, but low w_2 will amplify high-frequency noise and make the image too sharp with hammerstroke artifacts (**Figure E5(d, e)**). Users can adjust w_2 to achieve a balance between resolution and hammerstroke artifacts. In general, we set $w_1 = 0.5$ and $w_2 = 0.1$ to balance noise, artifact, and resolution. We suggest that for the image with a high signal-to-noise ratio, we can appropriately reduce w_2 to obtain a higher signal-to-noise ratio, and for the image with a low signal-to-noise ratio, we can appropriately increase w_1 to suppress noise.

What's more, we list **Table E2** to list of parameters of our results as shown in Table SN1. In general, we set w_1 default to 0.5 and w_2 default to 0.1 to balance noise, artifacts and resolution. And the parameters usually do not need users to adjust, which brings great convenience.

Figure E5 | The effect of different parameters on reconstructed results. a, the diagram of Filter1 and Filter2. **b,** the effect of different parameters (w_1 and w_2) on the profile of Filter1 and Filter2 (white line in a). **c,** the reconstructed results of the actin filament with $w_1 = 0.5$, $w_2 = 0.1$. **d,** effect of different w_2 on the reconstruction with **e,** the profile on the white line. **f,** Effect of different w_1 on the reconstruction. Scale bar: $2\mu\text{m}$. Scale on z-axis: 7 layers, $0.125\mu\text{m}$ per layer.

Action:

On Supplementary Note 1, explain the filters in detail.
 On Supplementary Note 4, added effect of parameters.

Q4: We also ask that you update the software to improve usability, along the lines of referee 1.

Response: We have made several optimizations to the Open-3DSIM in terms of improving user experience and increasing selectivity.

Figure E6 | The input and output methods after improvement. In the new version, **a**, users can drag input images into Fiji and then choose a certain input image in Fiji. **b**, the output images (WF, Open-3DSIM, and pSIM) will appear in Fiji automatically after reconstruction. In addition, we provide middle results for users to evaluate their data and system, including **c** the Fourier transform of raw data and **d** spectrum of separated bands.

As shown in **Fig. E6(a, b)**, now, users can select the input image in the image opened by Fiji. And the result will be automatically displayed in Fiji and the source file will be output under the input path (as you can see on our install video in Figshare: https://figshare.com/articles/dataset/Open_3DSIM_DATA/21731315).

At the same time, we added two extra raw data formats in three platforms, including the formats of the Nikon N-SIM system and home-built system, for users to choose from, and we displayed the multi-color imaging results of N-SIM in **Fig. 2e** of the revised manuscript. Then, we have added the middle results of reconstruction including the Fourier transform of raw data and separated bands for hardware specialists to check their data and home-built 3DSIM systems as shown in **Fig. E6(c, d)**. And we have added a typical dataset of Open-3DSIM with detailed parameters, reconstruction results, and comparisons in Figshare, Github and **Supplementary Note 12**.

What's more, we added the prompt of the running crash caused by an operation error. We give a prompt in the read data to the user who has not installed the MATLAB runtime correctly, as shown in **Fig E7(a)**. Then we will give a warning if the incorrect format of the input image is chosen in **Fig. E7(b)**. We also give a warning to the user who has insufficient RAM before the process data, as shown in **Fig. E7(c)**, and put forward a clear error message for the crash caused by insufficient RAM during the processing, as shown in **Fig E7(d)**. We hope that these improvements will further maximize the user-friendliness of Open-3DSIM to engineers and biologists and promote its further development.

Figure E7 | The error messages for **a**, MATLAB runtime is not correctly installed, **b**, The data format is not correct, **c**, The RAM may be out of memory, and **d**, The RAM is not enough.

Action:

On Manuscript, Page 6, Fig. 2(e), changed it to multi-color sample from N-SIM system;
On Supplementary Note 12, added the open-source dataset.
Improved the software.

Q5: We are not concerned with comments about "novelty", as we believe the tool to be of use to the community.

Response: Thank you for your support. We contribute to improving the performance and user-friendliness of the reconstruction of 3DSIM. First of all, we have proposed the adaptive parameter estimation based on +2nd and +1st frequency components, greatly improving the accuracy of estimation under low SNR. And the correlation methods without initial knowledge of parameters can greatly improve the compatibility with commercial or home-built 3DSIM systems. Then, we proposed two-step spectrum filters, which make the combined spectrum approach to the ideal, thus greatly reducing the artifacts and maintaining weak information.

What's more, we have proposed three platforms for Open-3DSIM to maximize its user-friendliness. We first develop GUI for 3DSIM in Fiji and Exe with three common input formats and can-be-chosen experimental/simulated OTF, and provide the middle results to evaluate the raw data, for convenience for developers and biologists.

Open-3DSIM is multi-platform, not limited by specific microscope workstations, and can be customized and adjusted according to user needs. And it is also extensible, it is fully compatible with other algorithm optimization methods based on regular terms or machine learning. Its superiority, universality, compatibility and user-friendliness can provide a solid software guarantee for the next development of 3DSIM.

Reviewer #1 (Remarks to the Author):

In their manuscript entitled “Open-3DSIM: an Open-source three-dimensional structured illumination microscopy reconstruction platform” authors Ruijie Cao et al. present both a variation and improvement of the original 3D SIM reconstruction algorithm (as introduced by Gustafsson et al. in 2008) as well as an open-source implementation of their improved algorithm.

Gustafsson’s approach of 3D SIM reconstruction, based on frequency-domain unmixing, that allows for a ‘direct’ reconstruction (the equation system is fully defined and can be analytically solved) is well established and widely used in commercial SIM applications. Improving on this algorithm, in case of the authors by modifying and tuning the filtering steps, is thus a welcome improvement, especially useful for low SNR data in live cell imaging. The approach presented (supplementary note S1 & S2) seems sound and their results (fig. 2, especially the comparison with the reconstructions provided by the OMX) make it clear that these improvements seem to work.

The manuscript is well written, both the mathematical foundation of 3D SIM reconstruction (supplementary note S1) as well as the algorithms data flow (supplementary note S2) are easy to follow and very helpful to understand the algorithms. The main manuscript, both text and especially figures, are densely packed with information and could offer some more discussion and explanation. However, I assume this is due to the ‘brief’ format chosen that sets upper limits on text length and number of figures, so it is not easily changed.

Together with improvements of the algorithm itself, the authors provide an open-source implementation of this algorithm, based on MATLAB and connected also to a Fiji plugin for easier integration into existing workflows. I view this software package, its open-source nature, general applicability to SIM data and also ease of use, as a core part of this publication.

As the authors state in the introduction, the SIM community is in need of an open-source, well established 3D reconstruction solution of 3-beam, 3D SIM datasets (as acquired on systems like the OMX, the Zeiss Elyra SIM or the Nikon N-SIM). I fully agree with this statement and am happy that they put effort into the work towards such a solution. In general, I fully support publication of the manuscript. However, in my opinion, some important points need to be addressed both in the manuscript and especially with the software and its packaging:

Response: We thank Review #1 very much for your support and constructive suggestions. In 2DSIM field, a series of algorithms such as fairSIM and HiFi-SIM are devoted to improving the performance of reconstruction and integrated in Fiji/MATLAB with user-friendly GUI, enabling biologists to easily use 2DSIM for reconstruction. However, in the multi-layer 3DSIM field, there is a lack of a user-friendly and well-estimated platform. To address this need, Open-3DSIM is developed here. All of your suggestions are very important and helpful for us to improve the manuscript and software packages, and we have revised them in detail for your consideration.

Manuscript:

Q1: In the introduction, the authors claim there is currently no general implementation of a 3D SIM reconstruction software widely available. To my knowledge, this is true in terms of software packages fully accompanied by an academic publication. However, at least the following packages

are both freely available and regularly used by the SIM community, so the authors might want to mention them:

a) <https://github.com/scopetools/cudasirecon> A GPU-accelerated version of the original Gustafsson algorithms, as far as I know based on the same codebase.

Authored by Lin Shao and maintained (bugfixed, not necessarily feature improvements) by Talley Lambert. In active use by groups.

b) https://github.com/Knerlab/SIM_Reconstruction A python-based implementation of 2D and 3D SIM reconstruction.

There is also a fairSIM version capable of 3D reconstruction (<https://github.com/fairsim/fairsim/tree/develop-3D-SIM>), but it is arguably still in a beta state (and development has become somewhat dormant), so I would not mention it here.

Response: We are very grateful for your suggestion of the other open-source 3DSIM algorithms. We have carefully investigated these open-source 3DSIM algorithms, properly corrected the description in the manuscript, listed **Table R1.1** for comparison, and performed experimental comparisons. In **Table R1.1**, Cudasirecon is the first algorithm you mentioned, and AO-3DSIM is the second. We have add the discussion in **Manuscript** Page2: Line 11-16(yellow) and add **Supplementary Note 7** for the experimental comparisons.

Table R1.1. Comparison of open-source 3DSIM algorithms.

Algorithm	Platform	Property	Purpose	Performance	Universality		
					Input formats	Initial parameters	OTF
Open-3DSIM	MATLAB Fiji (GUI) Exe (GUI)	Spectrum optimized and multi-platform	Optimize performance and Maximize user-friendliness	Artifact-reduced Noise-control	Three (OMX, NSIM, Home-built)	No	Experimental /Simulated
Cudasirecon	C++	Creation of wiener-based 3DSIM	Prove 3DSIM principle	Classical	One (OMX)	Need	Experimental
AO-3DSIM	Python	Traditional wiener-based 3DSIM	Adaptive optics to reduce artifacts	No optimization on algorithms	One (Home-built)	Need	Experimental /Simulated
SIMnoise	MATLAB	Iterative optimization denoising	Reconstruct well under low SNR	Noise-control	Two (OMX, Zeiss)	Need	Experimental /Simulated
3DSIM (4BSIM)	MATLAB	Traditional wiener-based 3DSIM	To compare 3DSIM with 4BSIM	No optimization on algorithms	One (Home-built)	Need	Experimental

The table to compare open-source 3DSIM algorithms is shown in **Table R1.1**. It can be seen that we first developed GUI within Fiji and Exe with three normal input formats, making it easy for biologists to use and compatible with different hardware systems. Then, our solution doesn't need initial parameters (such as frequency, and angle of illumination pattern) and experimental OTF, making it fully compatible with commercial or home-built 3DSIM systems. Last, we use adaptive parameter estimation and spatial spectrum optimization to achieve the optimal fidelity reconstruction with reduced artifacts, controlled noise, removed defocus, and retained weak information.

According to **Table R1.1**, except that SIMnoise and Open-3DSIM did optimization on Wiener-based 3DSIM, Cudasirecon, AO-3DSIM, and 3DSIM (4BSIM) are completely based on

traditional Wiener-based 3DSIM for certain hardware purposes. So, it is the reason that our manuscript mainly compares Open-3DSIM with SIMnoise and commercial OMX systems. And in response to the suggestions, we add **Supplementary Note 7 (Fig. R1.1 here)** to do comparisons between Open-3DSIM, AO-3DSIM, and the 3DSIM part of 4BSIM (simplified as 4BSIM) using their well-adjusted reconstructions. It can be shown that Open-3DSIM outperforms traditional Wiener-based 3DSIM algorithms including AO-3DSIM and 4BSIM with fewer artifacts and can retain weak information. And in **Fig. 2(a-d, g)** and **Supplementary 8-9**, we demonstrate its superiority compared with SIMnoise and OMX.

What's more, Open-3DSIM is multi-platform, not limited by specific microscope workstations, and can be customized and adjusted according to user needs. And it is also extensible, it is fully compatible with other algorithm optimization methods based on regular terms or machine learning. Its superiority, universality, compatibility and user-friendliness can provide a solid software guarantee for the next development of 3DSIM.

Figure R1.1 | Comparison of traditional Wiener-based 3DSIM algorithms including AO-3DSIM and 4BSIM. a, the actin of α -TN4 lens epithelial cell, whose reconstruction is derived from AO-3DSIM. **b**, the membrane of sporulating *B. subtilis*, whose reconstruction is derived from 4BSIM. Scale bar: 2 μ m. Scale on z-axis: **a**, 26 layers, 0.2 μ m per layer, **b**, 32 layers, 0.125 μ m per layer.

Action:

On Manuscript, Page 2, Line 9-22, revised the description;
 On Manuscript, Page 3, Line 13-14, revised the description;
 On Manuscript, Page 4, Line 15-18, revised the description;
 On Manuscript, Page 8, Line 1-3, revised the description;
 On Supplementary Note 7, SI Page 18, Line 1-16, added the comparison.

Q2: The authors mention the improved resolution provided by their algorithm, and visually this is supported by fig. 1 and 2. However, this claim might benefit from a more quantitative analysis,

typically performed by applying Fourier ring correlation or image decorrelation analysis to the data, to get a quantitative resolution estimate. This should especially be performed for the comparison to the datasets reconstructed by the OMX-supplied software (SoftWORX, assumably).

Response: We thank you very much for your suggestions. As some of our images are reconstructed in low SNR, we used image decorrelation analysis to avoid noise and quantify the resolution of Open-3DSIM, OMX, and SIMnoise. The decorrelation analysis can be seen in **Fig. R1.2**, and the resolution table is listed in **Table 1.2**. It can be seen that Open-3DSIM achieves better resolution and can maintain more weak information than other algorithms. But we also want to stress that the spectrum of Open-3DSIM is still limited in the petal-shaped spectrum cutoff without RL deconvolution process to guarantee its high fidelity. The improvement of resolution is caused by the elimination of partial artifacts and background, as well as the reasonable proportion of redistribution between high-frequency and low-frequency spectrum, which is mainly reflected in the retention of weak signals but cannot provide additional high-frequency information beyond the petal-shaped spectrum cutoff.

Figure R1.2 | The resolution comparisons between SIMnoise, OMX, and Open-3DSI using the image decorrelation method. Reconstruction results are from a, Fig. 2(c), b, Fig. 2(d), c, Fig. SN8(a), d, Fig. SN9(b), and e, Fig. SN9(c).

Table 1.2 Resolution comparisons of different algorithms using image decorrelation.

Figure	OMX	SIMnoise	Open
Fig. 2(c)	/	118.8nm	109.16nm
Fig. 2(d)	131nm	/	105.88nm
Fig. SN9(a)	/	130.68nm	111.24nm
Fig. SN9(b)	97.68	/	90.96
Fig. SN9(c)	135nm	143.32nm	132.2nm

Action:

On Supplementary Note 9, SI Page 20, Line 13 - Page 21, Line 3, added the description.

Reference

1. Descloux. A, Größmayer. K, Radenovic. A. Parameter-free image resolution estimation based on decorrelation analysis. Nature methods **16**(9): 918-924 (2019).

Q3: The timelapse panel in fig. 2 seems to be cut off in my version of the manuscript, I do not know if this is intentional.

Response: The lower part of **Fig. 2(f)** is separated. We are sorry for causing your misunderstanding. We intentionally separate the mitochondrial information of the *xoy* plane and *xoz* plane. For this possible misunderstanding, we have added detailed identification in **Fig. 2(f)** (**Fig. R1.3(a)** here).

Action:

On Manuscript, Page 6, Fig. 2(f), added *xoy* and *xoz*.

Figure R1.3 | The cropped image in our manuscript **Fig. 2(f)**, denoting the *xoy* plane and *xoz* plane. Scale bar: 2 μm . Scale bar on *z*-axis: 6 layers, 0.125 μm per layer.

Software: Unfortunately, I had some trouble getting the provided software to work on my machine. I was in the end able to run the software and confirm it generally reconstructs 3D SIM images. I however did not perform in-depth testing, as I am not sure my changes to the code are correct.

Q4: The provided Fiji plugin starts a GUI to control OpenSIM from within Fiji, but does not integrate with Fiji. In particular, input and output images are files on disks, not linked to the image stacks opened in Fiji. This makes interactive work and especially automation quite complex.

Response: According to your suggestion, we have made several optimizations to the Open-3DSIM (Fiji) version in terms of improving user experience and increasing selectivity. As shown in **Fig. R 1.4(a, b)**, now users can select the input image in the image opened by Fiji. And the result will be automatically displayed in Fiji and the source file will be output under the input path (as you

can see on our installation screen-capture video in Figshare: https://figshare.com/articles/dataset/Open_3DSIM_DATA/21731315).

At the same time, we added two extra raw data formats, including the formats of the Nikon N-SIM system and a home-built system, for users to choose from, and we displayed the multi-color imaging results of N-SIM in Fig. 2e of the revised manuscript. What's more, we have added the middle results of reconstruction including the Fourier transform of raw data and separated bands for hardware specialists to check their data and home-built 3DSIM systems as shown in Fig. R 1.4(c, d). We hope that these improvements will further maximize the user-friendliness of Open-3DSIM to engineers and biologists and promote its further development.

Figure R1.4 | The input and output methods after improvement. In the new version, **a**, users can drag input images into Fiji and then choose a certain input image in Fiji. **b**, the output images (WF, Open-3DSIM, and pSIM) will appear in Fiji automatically after reconstruction. In addition, we provide middle results for users to evaluate their data and system, including **c** the Fourier transform of raw data and **d** spectrum of separated bands.

Action:

On Manuscript, Page 6, Fig. 2(e), changed it to multi-color sample from N-SIM system;

Improved the software.

Q5: On my machine, the plugin crashed w/o a clear error indication. I did not investigate this further, but am happy to do so with a revised version of the manuscript and code. With the technical realization chosen by the authors, there are however two points that can be addressed to make crashes much less likely:

a) Checking for available memory, and providing clear error messages for out-of-memory problems. In my experience, 3D SIM reconstructions are demanding in terms of memory, so lower-end machines (8GB, even 16GB of RAM) will often run out of memory.

b) Since bundling with the MATLAB runtime is used, checking the code on a machine where no MATLAB is installed, and only the install guide is followed (easy to do in a virtual machine that can be reset). This avoids the problem of libraries missing on machines that do not have MATLAB installed.

Response: We are very grateful for the valuable suggestion. To address this bug, we have provided a detailed Fiji installation video with more 3DSIM data on the Figshare website (https://figshare.com/articles/dataset/Open_3DSIM_DATA/21731315). We have tested it on over 10 computers and the tests are all successful.

As we cannot repeat the same bug scenario, we added the prompt of the running crash caused by an operation error. We give a prompt in the read data to the user who has not installed the MATLAB runtime correctly, as shown in **Fig R1.5(a)**. Then we will give a warning if the incorrect format of the input image is chosen in **Fig. R1.5(b)**. We also give a warning to the user who has insufficient RAM before the process data, as shown in **Fig. R1.5(c)**, and put forward a clear error message for the crash caused by insufficient RAM during the processing, as shown in **Fig R1.5(d)**.

Figure R1.5 | The error messages for **a**, MATLAB runtime is not correctly installed, **b**, The data format is not correct, **c**, The RAM may be out of memory, and **d**, The RAM is not enough.

Action:

Improved the software.

Q6: The actual implementation of the algorithm is provided in MATLAB. Both versions (the one provided with the review material and the one on github) crashed with a missing reference to the ‘phase’ function in the fit_peak and find_illumination routines. Changing the ‘phase’ to the ‘angle’ function here (which seems to be the value to be retrieved here) got me to a working version of the code, and reconstructions look reasonable. However, I would of course like to cross-check with the authors if this is the correct fix for this bug, and how it occurred.

Response: We are sorry for this problem, and your modification for this bug is very correct. The “angle” function and the “phase” function have no difference to solve the angle of a single complex number in our code. We tested MATLAB 2018~2022 versions and found that the 2019a and 2019b do not support the use of the “phase” function, but all versions support the use of “angle”. Therefore, we carefully changed all the “phase” functions to the “angle” function, and package it with the revised manuscript.

Action:

Improved the software.

Q7: Software repositories like *github* typically contain source code directly, so changes can be tracked and collaboration on code is possible. The repository provided by the authors, however, contains compressed archives (rar and zip), that then contain a copy of the code each. I would recommend changing this to a more standard format, where the repository contains the pure source code. To provide binary distributions (like zip files that are convenient for the end user), github offers the ‘release’ feature, where such data can be attached to (each version of) the source code.

Response: We sincerely thank reviewer#1 for this helpful suggestion. We have reorganized the Github repository and tags as shown in **Fig. R1.6**. Now, the repository only contains the source-code of three versions including MATLAB, Java, and Exe. And the v2.0 version is uploaded into the Tags using “release” feature into Github.

Figure R1.6 | The new Github version of Open-3DSIM. **a**, now, the repository only contains the source-code of three versions including MATLAB, Java, and Exe. **b**, the v2.0 version is uploaded into the Tags using “release” feature into Github.

Action:

Improved the Github repository.

Q8: There seems to be no clear license attached to the source code. As the authors aim to provide an open-source solution, I would recommend assigning one of the standard open-source software licenses (GPL, BSD-2, ...) to their code. This should be stated both in the manuscript, in the source code repository and the distributed binary files.

Response: We sincerely thank reviewer#1 for this helpful suggestion. We claim an Apache License for Open-3DSIM to make it open-source, and can be used/modified in commercial close-source developments. We have added the relevant license information to the manuscript, as well as three platforms (MATLAB, Fiji, and Exe), Github, Figshare, and the binary files.

Action:

On Manuscript, Page 11, Line 6-7, claimed an Apache liscence.

Reviewer #2 (Remarks to the Author):

The paper by RuiJie Cao et al. open-sourced a 3DSIM super-resolution image reconstruction algorithm, named Open-3DSIM. Although there are several commercial 3D-SIM instruments, such as GE OMX, Nikon N-SIM, Zeiss Elyra-SIM, their image reconstruction remained a black box for SIM users and researchers. Like the first open-source 2DSIM published in Nature Comm. in 2016 which triggered many follow-up studies to optimize the performance of reconstruction algorithms, I expect Open-3DSIM will have similar impacts in the field and eventually boost the application of 3DSIM in cell biology.

Open-3DSIM algorithm is based on the established principle of 3DSIM. The authors adapted spectral filtering to diminish the artifacts. One novelty is the extension of 3DSIM to polarization-SIM so that SIM in six dimensions ($xyz\theta\lambda t$) can be realized. The demonstrated reconstruction results outperform the commercial 3D SIM and previously published SIMnoise. The supplementary code is complete, well-understandable, and easy to follow.

There are some issues that the authors should be addressed to make the paper more rigorous:

Response: We thank Review #2 very much for your support and suggestions. In our research, we find that the SIM field is in lacks of a well-established and user-friendly 3DSIM algorithm, so we have developed Open-3DSIM. We hope Open-3DSIM will contribute to this community and make 3DSIM easy to access for developers and biologists. We have carefully corrected our manuscript and answered your questions in detail for your consideration.

Q1: Since Open-3DSIM is designed for multi-layer 3D SIM and the advantage of multi-layer 3D SIM over single-layer is resolution doubling in z-direction. So the authors should give a quantitative resolution calibration using fluorescent beads or 3D patterns in Argolight. The current paper only shows decreased defocused backgrounds and artifacts without quantification.

Response: We sincerely thank you for your suggestion. We have added **Supplementary Note 5** to quantitatively explain the improvement of the z-axis resolution of multi-layer 3D SIM compared with single-layer 3DSIM. As shown in **Fig. R2.1 (a) and (b)**, taking the Argolight pentagon pattern and 100 nm fluorescent beads of 488nm excitation wavelength as examples, we take 10 spatial points on Argolight and beads to make the intensity-pixel curve (processing with spline interpolation) on the z-axis as shown in **Fig. R2.1 (c) and (d)**.

We used the full-width at half-maximum (FWHM) analysis function of origin to obtain the half-height full-size of different algorithms of Arglight and Beads. It can be seen that the half-height and full-width on the z-axis of WF, single-layer 3D SIM, and multi-layer 3D SIM are 1218nm, 1412nm, and 765nm for Argolight, respectively. And the half-height and full-width on the z-axis of WF, single-layer 3D SIM, and multi-layer 3D SIM are 596nm, 525nm, and 387nm for beads, respectively.

At the same time, we use the PSFj plugin to quantitatively analyze the resolution of beads, and get that the resolution of the xoy plane and z-axis of WF is 251nm, 717nm, the resolution of the xoy plane and z-axis of single-layer 3DSIM is 120nm, 708nm, and the resolution of the xoy plane and z-axis of multi-layer 3DSIM is 118nm, 344nm. Therefore, from a quantitative point of

view, we show that 3DSIM achieves double 3D resolution compared with WF and double z-resolution compared with single-layer 3DSIM.

Figure R2.1 | Quantitative resolution calibration of WF, HiFi-SIM, and Open-3DSIM, including results of **a**, Argolight, and **b**, sparse fluorescent beads. Scale bar: 2 μ m. Scale on z-axis: **a**, 8 layers, **b**, 33 layers, 0.125 μ m per layer.

Action:

On Supplementary Note 5, SI Page 14, Line 6-9, added reconstruction of Argolight and beads.

On Supplementary Note 5, SI Page 15, Line 4-21, added z-axis resolution qualification.

Q2: Open-3D SIM used two-step frequency domain filtering to reduce the artifacts and improve the resolution. The filtering includes several designed functions and parameters. It is expected that different parameter settings have clear effects on the final results. So, the authors may a table to list the parameters used in the demonstrated results and recommend how to set the parameters.

Response: Thanks for your suggestions, we have added **Supplementary Note 4** to guide users to choose the parameters and list a table of **Table R2.1** to list the parameters in all our reconstructions. In general, parameters in Open-3DSIM are usually fixed and easy to be adjusted, which brings convenience for users.

In our work, we set all reconstructions with edeletaper (edge smoothing) and attenuation (*att* in eq(9)) equal to 0. To preserve edge and high-frequency information, we also recommend users keep the default settings. However, for the original image with lots of edge information, a reasonable set of some Edeletaper (typically 1-10 pixels for 512×512 images) can help to reduce the artifacts; In the case of serious high-frequency noise, a reasonable set of Attenuation between 0 and 0.5 will help to reduce the noise. Notchwidth1(*w* to construct OTF_{notch}) and Notchwidth2(*w* to construct notch-filter) are notch range parameters related to image size, and Notchdepth1(*d* to construct OTF_{notch}) and Notchdepth2(*d* to construct notch-filter) are constant notch depth. All reconstructions in this article maintain the default parameters of these four variables, and users are also recommended to keep the default parameters. However, in the process of spectrum analysis after reconstruction, if honeycomb artifacts caused by an insufficient notch (obvious high-frequency modulation point) are found, Notchdepth2 and Notchwidth2 can be appropriately increased, and Notchdepth1 and Notchwidth1 can be appropriately reduced.

To the users, we set w_1 and w_2 to help users to adjust parameters. As shown in **Fig. R2.2(a)**, Filter1 generally presents a low-pass filter while Filter2 presents a high-pass filter. Larger w_1 will make Filter1 more obvious in suppressing the high-frequency information in the OTF of each spectrum, and smaller w_2 will enhance the high-frequency part of the petal-shaped spectrum (**Fig. R2.2 (b)**).

We also use the actin filament in **Fig. SN3** as an example, and the well-adjusted parameter is $w_1 = 0.5$ and $w_2 = 0.1$ in **Fig. R2.2(c)**. Filter1 is mainly used for spectrum editing after abnormal reorganization in cooperation with the previous notch-filter. When w_1 is larger, the suppression of high-frequency noise at various spectral levels is more obvious, and the ability to suppress sidelobe artifacts is enhanced (**Fig. R2.2 (f)**). But the high-frequency information may be suppressed too. Filter2 is mainly used to preserve weak signals, but low w_2 will amplify high-frequency noise and make the image too sharp with hammerstroke artifacts (**Fig. R2.2 (d, e)**). Users can adjust w_2 to achieve a balance between resolution and hammerstroke artifacts. We suggest that for the image with a high SNR, we can appropriately reduce w_2 to magnify high-frequency information, and for the image with a low signal-to-noise ratio, we can appropriately increase w_1 to suppress noise and artifact.

What's more, we list a table to list of parameters of our results as shown in **Table SN1**. In general, we set $w_1 = 0.5$ and $w_2 = 0.1$ to balance noise and artifacts and retain weak information. And the parameters usually do not need users to adjust, which brings great convenience.

Table R2.1 The parameters and data origin of all of the reconstructions in our work.

Figure	Exwavelength(nm)	w1	w2	Data origin
Fig. 1(c)	488	0.5	0.1	OMX
Fig. 2(a)	568	0.5	0.1	OMX
Fig. 2(c)	488	0.5	0.1	SIMnoise
Fig. 2(d)	488	0.5	0.1	OMX
Fig. 2(e)	488, 561, 609	0.5,0.5,0.8	0.1,0.1,0.1	N-SIM
Fig. 2(f)	647	0.5	0.1	OMX
Fig. 2(g)	568	0.5	0.1	OMX
Fig. SN3(a, b)	488	0.5	0.1	fairSIM, OMX
Fig. SN4	488	0.8	0.1	fairSIM
Fig. SN5(a, b)	488,488	0.5,0.5	0.1,0.1	OMX, fairSIM
Fig. SN5(c, d)	488,488	0.5,0.5	0.1,0.1	OMX, OMX
Fig. SN6(a, b)	488	0.5	0.1	Simulation
Fig. SN7(a, b)	647,488	0.5,0.5	0.1,0.1	AO-3DSIM, 4BSIM
Fig. SN8(a, b)	488,488	0.5,0.5	0.1,0.1	OMX
Fig. SN9(a, b, c)	488,568,568	0.5,0.5,0.5	0.02,0.1,0.1	SIMnoise, OMX, OMX
Fig. SN10	405,488,568	0.5,0.5,0.5	0.1,0.1,0.1	OMX
Fig. SN11	488	0.5	0.1	OMX

Figure R2.2 | The effect of different parameters on reconstructed results. a, the diagram of Filter1 and Filter2. **b**, the effect of different parameters (w_1 and w_2) on the profile of Filter1 and Filter2 (white line in a). **c**, the reconstructed results of the actin filament with $w_1 = 0.5$, $w_2 = 0.1$. **d**, effect of different w_2 on the reconstruction with **e**, the profile on the white line. **f**, Effect of different w_1 on the reconstruction. Scale bar: $2\mu\text{m}$. Scale on z-axis: 7 layers, $0.125\mu\text{m}$ per layer.

Action:

On Supplementary Note 4, added effect of parameters.

Q3: The scale bar for z-direction is not shown for all x-z section images. Is that the same for x-y direction? Generally, it is different.

Response: We sincerely apologize for not classifying the x-z scale bar and it is not the same for the x-y direction. We have described it in all figure legends in our revised manuscript in the form of: Scale on the z-axis: 41 layers, 0.125 μ m per layer.

Action:

Revised all the figure legends.

Q4: There are some errors in the paper. For example, in line 122 of the main text, it should be "Supplementary note 4" instead of "Supplementary note 3".

Response: We feel very sorry for the mistakes. We have carefully revised the questions and checked the full text in detail.

Q5: In supplementary data, only one set of 3D image data is included. It is recommended that the authors also include other datasets, for example, data for Argolight and data for polarization 3DSIM.

Response: We sincerely agree with your proposal and feel sorry for not providing it in time. After submitting the first draft, we uploaded the typical 3DSIM raw data, parameter setting file, and comparison data with other algorithms (**Table R2.2** here, **Supplementary Note 12** in manuscript) to Figshare(https://figshare.com/articles/dataset/Open_3DSIM_DATA/21731315) for the reference of reviewers and users, and copied the link to our GitHub (<https://github.com/Cao-ruijie/Open3DSIM>) and manuscript. Each data has very detailed parameter settings. We think that these data and comparisons will provide strong support for our work.

Table R2.2 The open-source file list.

File	Explanation	Exwavelength	Oil	Image size
OMX_Argolight_ex488_oil1518(253M).dv	OMX raw data	488nm	1.158	512×512×33
OMX_COS7_Nup_ex568_oil1514(192M).dv	OMX raw data	568nm	1.514	512×512×25
OMX_Mouse_section_ex568_oil1518(315M).dv	OMX raw data	568nm	1.518	512×512×41
OMX_U2OS_Actin_ex488_oil1518(77M).dv	OMX raw data	488nm	1.518	512×512×10
OMX_U2OS_Actin_ex568_oil1518(100M).dv	OMX raw data	568nm	1.512	512×512×13
SIMnoise_Beads_ex488_oil1512(63M).dv	SIMnoise raw data	488nm	1.512	512×512×33
SIMnoise_C127_Tubulin_ex488_oil1512(79M).dv	SIMnoise raw data	488nm	1.512	512×512×41
NSIM_U2OS_Actin_ex488_oil1512(83M).tif	NSIM raw data	488nm	1.512	1024×1024×11
Comparison	Reconstructed results and comparisons	/	/	/
Install_Fiji_Screenshot.mp4	Guide vedio to install Fiji version	/	/	/
Parameter.zip	Paramters, OTFs	/	/	/

Action:

On Supplementary Note 12, added the open-source dataset.

Uploaded dataset into

Figshare:

https://figshare.com/articles/dataset/Open_3DSIM_DATA/21731315

Q6: In the Algorithm flow (Supplementary Note2), it shows three images of phase 0, phase 1, and phase 2. To remove the confusion, the authors may draw this figure with 5 images of phases 0-4 and also include the separate bands of -1,-2 order.

Response: We sincerely agree with your suggestions, and have revised them in **Supplementary Note 2** and **Figure R2.3** for your consideration.

Figure R2.3 | Algorithm flow and intermediate operations in frequency or spatial

domain.

Action:

On Supplementary Note 2, improved the algorithm flow and description.

Reviewer #3 (Remarks to the Author):

Remarks to the Author:

This manuscript describes an open-source software package that can process 3D-SIM raw data to produce super-resolved images with the additional dipole-orientation info. While the package could in time prove to be a very useful toolkit for 3D-SIM community and the effort is much applaudable, I don't recommend its publication as a research article or brief in Nature Methods based on the following reasoning:

Response: We sincerely thank Reviewer#3 very much for the detailed and helpful comments. These comments significantly improved the quality of our paper, and we have revised them in our new manuscript, which we sincerely ask for your reconsideration.

Q1. Lack of novelty.

Q1a. There is no significant breakthrough in the 3D-SIM reconstruction algorithm. What the authors proposed, as far as I can understand from the suppl notes, are not that different from the original Gustafsson et al 2008 paper. In fact, there might be some mistakes in SN1 about those filters and more on those below.

Response: We admit that the principle of Open-3DSIM does not have a significant breakthrough over Gustafsson's 2008 paper. But we present both a variation and improvement of the original 3D SIM reconstruction algorithm with improved performance and user-friendliness. First of all, we have proposed the adaptive parameter estimation based on +2nd and +1st frequency components, greatly improving the accuracy of estimation under low SNR. And the correlation methods without initial knowledge of parameters can greatly improve the compatibility with commercial or home-built 3DSIM systems. Then, we proposed two-step spectrum filtering, which makes the combined spectrum approach close to the ideal spectrum, thus greatly reducing the artifacts and maintaining weak information. We have compared our results with other non-open-source commercial platforms such as GE DeltaVision OMX or open-source algorithms such as SIMnoise with superior performance.

What's more, we have proposed three platforms for Open-3DSIM to maximize its user-friendliness. We first develop GUI for 3DSIM in Fiji and Exe with three common input formats, without prior knowledge of illumination pattern, can-be-chosen experimental/simulated OTF, and provide the middle results to evaluate the raw data, for convenience for developers and biologists.

To the mistakes, we have corrected the mistake in **Fig. SN1(a)**. And we have presented a detailed explanation of our two-step filtering process.

Q1b. The polarization or dipole-orientation analysis included in the package is completely based on a previously published technique

Response: In this work, we quoted the work of "Super-resolution imaging of fluorescent dipoles via polarized structured illumination microscopy" in Nature communications in 2019.

Although users can obtain polarization analysis using the Matlab code we provided in Nature communications 2019, the code is intended to show the principle of a hyper-space polarization structured illumination microscopy analysis. The code also does NOT contain 3DSIM

reconstruction and its 3DSIM reconstruction is completely based on OMX reconstruction. Consequently, it is inconvenient for users to obtain polarization data because of the need to input the original data, illumination frequency, angle, and other reconstruction parameters, as well as the reconstructed data.

The major role of Open-3DSIM is to perform the 3D-SIM reconstruction in an open-source, user-friendly platform. After the reconstruction, users only need to click one button to obtain the polarization information of the fluorescence image.

In **Fig. 2g** (or **Fig R3.1**), we compared the polarization orientation results based on Open-3DSIM, SIMnoise, and OMX, and it can be seen that Open-3DSIM has a superior capability of artifact removal and defocus elimination, which has a great contribution to the accurate acquisition of polarization information.

Figure R3.1 | Algorithm flow and intermediate operations in frequency or spatial domain. Scale bar: 2 μ m. Scale on z-axis: 7 layers, 0.125 μ m per layer.

Q1c. The authors try to sell their package as the only 3D-SIM open source package available. This is not true; there is the "cudasirecon" package downloadable from Github: <https://github.com/scopetools/cudasirecon>

Response: We are very grateful for this helpful suggestion, which is also mentioned by Reviewer 1. We have carefully investigated the open-source 3DSIM algorithms, properly corrected the description in the manuscript, listed **Table R3.1** for comparison, and done experimental comparisons. We have added the discussion in **Manuscript** Page2: Line 11-16(yellow) and added Supplementary Note 7 for the experimental comparisons.

The table to compare open-source 3DSIM algorithms is shown in **Table R3.1**. It can be seen that we first developed GUI within Fiji and Exe with three normal input formats, making it easy for biologists to use and compatible with different hardware systems. Then, our solution doesn't need initial parameters (such as frequency, and angle of illumination pattern) and experimental OTF, making it fully compatible with commercial or home-built 3DSIM systems. Last, we use adaptive parameter estimation and spatial spectrum optimization to achieve the optimal fidelity reconstruction with reduced artifacts, controlled noise, removed defocus, and retained weak information.

Table R3.1. Comparison of open-source 3DSIM algorithms.

Algorithm	Platform	Property	Purpose	Performance	Universality		
					Input formats	Initial parameters	OTF
Open-3DSIM	MATLAB Fiji (GUI) Exe (GUI)	Spectrum optimized and multi-platform	Optimize performance and Maximize user-friendliness	Artifact-reduced Noise-control	Three (OMX, NSIM, Home-built)	No	Experimental /Simulated
Cudasirecon	C++	Creation of wiener-based 3DSIM	Prove 3DSIM principle	Classical	One (OMX)	Need	Experimental
AO-3DSIM	Python	Traditional wiener-based 3DSIM	Adaptive optics to reduce artifacts	No optimization on algorithms	One (Home-built)	Need	Experimental /Simulated
SIMnoise	MATLAB	Iterative optimization denoising	Reconstruct well under low SNR	Noise-control	Two (OMX, Zeiss)	Need	Experimental /Simulated
3DSIM (4BSIM)	MATLAB	Traditional wiener-based 3DSIM	To compare 3DSIM with 4BSIM	No optimization on algorithms	One (Home-built)	Need	Experimental

According to **Table R3.1**, except that SIMnoise and Open-3DSIM did optimization on Wiener-based 3DSIM, Cudasirecon, AO-3DSIM, and 3DSIM (4BSIM) are completely based on traditional Wiener 3DSIM for certain hardware purposes. So, it is the reason that our manuscript mainly compares Open-3DSIM with SIMnoise and commercial OMX systems. And we add **Supplementary Note 7 (Fig. R3.2** here) to do comparisons between Open-3DSIM, AO-3DSIM, and the 3DSIM part of 4BSIM (simplified as 4BSIM) using their standard sample data with their adjustment on parameters. It can be shown that Open-3DSIM outperforms traditional Wiener-based Gustafsson-3DSIM algorithms including AO-3DSIM and 4BSIM with fewer artifacts and can retain weak information. And in **Fig. 2(a-d, g)** and **Supplementary 8-9**, we demonstrate its superiority compared with SIMnoise and OMX.

What's more, Open-3DSIM is multi-platform, not limited by specific microscope workstations, and can be customized and adjusted according to user needs. And it is also extensible, it is fully compatible with other algorithm optimization methods based on regular terms or machine learning. Its superiority, universality, compatibility and user-friendliness can provide

a solid software guarantee for the future development of 3DSIM.

Figure R3.2 | Comparison of traditional Wiener-based 3DSIM algorithms including AO-3DSIM and 4BSIM. **a**, the actin of α -TN4 lens epithelial cell, whose reconstruction is derived from AO-3DSIM. **b**, the membrane of sporulating *B. subtilis*, whose reconstruction is derived from 4BSIM. Scale bar: 2 μm . Scale on z-axis: **a**, 26 layers, 0.2 μm per layer, **b**, 32 layers, 0.125 μm per layer.

Action:

On Manuscript, Page 2, Line 9-22, revised the description;
 On Manuscript, Page 3, Line 13-14, revised the description;
 On Manuscript, Page 4, Line 15-18, revised the description;
 On Manuscript, Page 8, Line 1-3, revised the description;
 On Supplementary Note 7, SI Page 18, Line 1-16, added the comparison.

Q1d. The comparison between 3D-SIM processing results of this package with slice-by-slice 2D-SIM processing results from other published packages mostly misses the point: 3D-SIM data of course should be processed as a whole in 3D, as established in the original 3D-SIM paper, and of course it should outperform slice-by-slice 2D-SIM processing results. Although the latter's application in 3D-SIM can be attempted, its main value lies in such applications as single-layer 2D-SIM imaging without the benefit of TIRF configuration or very thin-layered samples. And it's not meant to compete with 3D-SIM in 3D performances.

Response: We agree with Reviewer#3. The multi-layer processing method should outperform the single-layer processing method, which is indeed not comparable. We agree that 3D-SIM data should be processed as a whole in 3D.

But as a counter-example, many existing SIM reconstruction software packages such as fairSIM and HiFiSIM do use multi-layer 3DSIM data for single-layer processing. And, it is noteworthy that the role of single-layer 3DSIM in this paper is only as a reference to show the improvement of z-axis resolution. In response to the reviewer's suggestion, we modified the

description of the comparison between single-layer 3DSIM and multi-layer 3DSIM at the end of **Supplementary Note 5**: We would like to stress that our comparison here is to illustrate the advantages of multi-layer 3DSIM in improving z-axis resolution compared with single-layer 3DSIM. 2DSIM (or single-layer 3DSIM) imaging has lower phototoxicity and faster imaging speed. On the contrary, multi-layer 3DSIM imaging has the ability of whole-cell imaging with resolution improvement on the z-axis. Users should reasonably select the required imaging method according to their needs.

Action:

On Supplementary Note 5, SI Page 15, Line 17-21, explained caveats about comparing the 3DSIM reconstructions with 2D reconstructions.

Q2. Doubt-sowing mistakes

Q2a. Line 17-18 Page 3: the authors wrote "the modulation depth will rapidly decrease with the bias of the focal plane". Even this sentence has little to do the focus of the manuscript, it is nevertheless incorrect (especially so in the related Fig SN1(a), lower half) and thus gives me excuses to doubt the merit of the manuscript. The modulation depth should remain ~constant with defocus. The authors might be referring to the gradual damping of the axial components of the SIM pattern, as seen in Fig 3(b) of the Gustafsson et al 2008 paper. That damping effect was from the spatial incoherence introduced by the multi-mode fiber in that particular SIM setup. Regardless of coherent or incoherent light source, the lateral modulation depth should be more or less constant with defocus. What's worse, Fig SN1(a) depicts an axially confined SIM pattern, which cannot be farther from reality.

Response: We are sincerely sorry for this error. It can be seen in eq (1) in **Supplementary Note 1** that the illumination pattern $I_{\theta,\varphi}(\mathbf{x}, \mathbf{y}, \mathbf{z})$ can be expressed as:

$$I_{\theta,\varphi}(\mathbf{x}, \mathbf{y}, \mathbf{z}) = I_0 |1 + 2m \cdot e^{j2\pi p_z z} \cdot \cos[2\pi p_{xy}(\cos\theta \cdot \mathbf{x} + \sin\theta \cdot \mathbf{y}) + \varphi]|^2$$

Where m denotes the modulation depth of the focal plane. It can be seen that m remains a constant with different z regardless of the coherent or incoherent light source.

What we originally drew in Fig. SN1(a) is the $I_{\theta,\varphi}(\mathbf{r}, \mathbf{z}) \otimes H(\mathbf{r}, \mathbf{z})$ using an ideal OTF. But this expression is wrong indeed. In response to your suggestion, we have revised **Fig. SN1(a)** to express $I_{\theta,\varphi}(\mathbf{r}, \mathbf{z})$. The revised **Fig. SN1** is shown in **Figure R3.3**.

Figure R3.3 | Principle of 3DSIM. **a**, The intensity distribution in the xoy and xoz plane. **b**, Shifted 3D frequency domain fills the leaky cone of OTF and **c** doubles the spectrum range, with **d** the further expansion of polarization dimension (red color). Purple, green, blue, and red color represent 0^{th} , $\pm 1^{\text{st}}$, and $\pm 2^{\text{nd}}$ frequency components, and polarization, respectively.

Action:

On Manuscript, Page 3, Line 4-5, corrected description.

On Supplementary Note 1, Fig. SN1(a), corrected the intensity distribution on the xoz plane.

Q2b. From what's written in SN1 about the series of filters, I don't see a theoretical basis that supports the plausibility of the algorithm. It seems that both Filter1 and Filter2 contain the OTF in the denominator, and wouldn't that mean the after applying them in series, the raw image would have been inverse-filtered twice and therefore the result be wrong? In SN3, related the spectrum filtering, why is the comparison with "Raw" and not with results from other 3D-SIM processing methods?

Response: Thank you for your suggestions. We have revised **Supplementary Note 1** to make the theory clearer and add the comparison between different 3D-SIM methods in **Supplementary Note 3**. We have the following response for your reconsideration.

(1) Theoretical basis in summary:

We have revised **Supplementary Note 1** to make the theory clearer. And state a simplified explanation here. In SIM, due to the abnormal spectrum distribution of the combination, various artifacts are usually caused. One typical problem is honeycomb artifacts caused by the high-frequency peak, which is caused by the abnormal distribution of $C_{SR,0}(k)$ with six frequency peaks of $\pm 2^{\text{nd}}$ components and six frequency peaks of $\pm 1^{\text{st}}$ components. So, the first optimization of Open-3DSIM is to use a designed notch-filter to suppress the high-frequency peaks. It is

noteworthy that the ideal spectrum of 3DSIM is the same form of wide-field, being smooth and even in the 3D domain. We think the ideal 3D spectrum is smooth and petal-shaped to be approached, and the distribution of directly combined spectrum after the notch-filter is like the sum of $OTF \cdot notch$. So, the first filter is described as $Filter1(k) = \frac{Apo}{OTF_{notch+w_1^2}}$ to make the spectrum approach to the ideal one. As shown in **Fig SN3**, applying $Filter1(k)$ can greatly suppress the patchy features, high-frequency noise, so as to suppress sidelobe artifacts, but the resolution is decreased because of the suppress of the edge of the petal-shaped spectrum. As a result, some weak information may be disappeared. What's more, with the involvement of w_1 , the approach to the ideal spectrum and the compensation for the previous notch-filter is not complete. So, we adapt an extra filter called $Filter2(k) = \frac{Apo}{OTF_{notch+w_2^2}}$ to continuously approach the ideal spectrum and increase the resolution of the reconstructed image, and the proportion of high-frequency component can be properly increased. $Filter2(k)$ will greatly maintain the weak information. The cooperation of two filters can fill the spectrum hole caused by the previous notch-filter and enhance the smoothness of the 3D spectrum.

(2) Theoretical basis in detail:

In the reconstruction of SIM, there are several typical problems: (1) traditional parameter estimation may make an error under low SNR or low modulation depth^{1,2}; (2) The peak of high frequency componets may result in honeycomb artifacts¹; (3) The abnormal spectrum may cause sidelobe artifacts³; (4) The noise in high-frequency part may cause hammerstroke artifacts¹; and (5) The traditional blur of high-frequency part will decrease the weak information in sample⁴.

To solve these problems, we firstly proposed an adaptive parameter estimation method as shown in **Fig. 1(a)**. This method can greatly improve the correctness of parameter estimation under low SNR using both 1st and 2nd frequency componets.

Then, to reduce the honeycomb artifacts (typical honeycomb artifacts can be seen in the OMX result in **Fig. SN3(b)**), the first spectrum optimization of Open-3DSIM is to use a notch-filter to suppress the high-frequency peaks.

$$OTF^{att}(x, y, z, n) \mathbf{C}_{SR_1}(k) = \sum_{n=-2}^2 \mathbf{C}_{ns}(k) \cdot notch(x, y, z, n) \cdot \quad (9)$$

Where $notch(x, y, z, 0) = 1 - d \cdot \exp\left[\left(\frac{x^2+y^2}{|p_{x,y}|^2} + \frac{z^2}{|p_z|^2}\right)/2/w^2\right]$ is designed according to the estimated frequency on the xoy and yoz plane, $notch(x, y, z, n)$ is the corresponding shift in frequency-domain position on the base of $notch(x, y, z, 0)$, $OTF(x, y, z, n)$ is the leaky-cone-shaped optical transfer function (OTF) shifted to the n -th place in the frequency domain, att is the frequency attenuation, and d and w are the notch depth and width respectively. To make the notch-filter applicable to samples with different layers and different wavelengths, the notch-filter is designed according to $p_{x,y}$ and p_z , which will greatly improve the universality and user-friendliness of Open-3DSIM with a fixed preset d and w .

However, as shown in **Fig. SN3(a)**, the combination of notched frequency components will

still cause sidelobe artifacts because of the patchy features in the combined frequency domain. And the excessive notch will cause the loss of high-frequency signal, so we designed a spatial sum of $OTF \cdot notch$ as OTF_{notch} to design the filter for spectrum optimization.

$$OTF_{notch} = \sum_{n=-2}^2 m(n) \cdot OTF(x, y, z, n) \cdot notch(x, y, z, n) \quad (10)$$

Where $m(n)$ is the weight coefficient of different Fourier orders. It is noteworthy that the ideal spectrum of 3DSIM is smooth and even in the 3D domain. We think combined OTF (petal-shaped) is the ideal spectrum to be approached, and the directly combined notched spectrum is like the distribution of OTF_{notch} . Thus, we adopt the first filter called $Filter1(k) = \frac{Apo}{OTF_{notch} + w_1^2}$ to correct the abnormal frequency $C_{SR_1}(k)$ to $C_{SR_2}(k)$.

$$C_{SR_2}(k) = C_{SR_1}(k) \cdot \frac{Apo}{OTF_{notch} + w_1^2} \quad (11)$$

Where Apo is the apodization function in the 3D frequency domain, w_1 is the parameter to design $Filter1(k)$. As shown in **Fig SN3(a)**, applying $Filter1(k)$ can greatly suppress the patchy features and high-frequency noise, so the ability to suppress artifacts is enhanced, but weak information is decreased because of the reduction in the edge of the petal-shaped spectrum. As a result, some weak information may be disappeared. What's more, with the involvement of w_1 , the approach to the ideal spectrum and the compensation for the previous notch-filter is not complete. So, we adapt an extra filter called $Filter2(k) = \frac{Apo}{OTF_{notch} + w_2^2}$ to continuously approach the ideal spectrum and retain weak information of the reconstructed image. The final spectrum $C_{SR_3}(k)$ can be expressed as:

$$C_{SR_3}(k) = C_{SR_2}(k) \cdot \frac{Apo}{OTF_{notch} + w_2^2} \quad (12)$$

The format of $Filter2(k)$ is the same as $Filter1(k)$, but with a relatively smaller w_2 (**Fig. SN4(b)**), the proportion of high-frequency component can be increased. $Filter2(k)$ will greatly maintain the weak information. The cooperation of two filters of $C_{SR_1}(k)$ and $C_{SR_2}(k)$ can fill the spectrum hole caused by the previous notch operation and enhance the smoothness of the SIM spectrum. The proper selection of w_2 can greatly maintain the weak information with no observable hammerstroke artifacts as shown in **Supplementary Note 4**. And The effect of Filter1, Filter2, and the comparison of the spectrum between different algorithms can be seen in **Supplementary Note 3**.

So, the final super-resolution 3DSIM image I_{SR} can be finally expressed as:

$$I_{SR} = F^{-1}[C_{SR_3}(k)] \quad (13)$$

(3) Correctness of Filter1 and Filter2:

Firstly, Filter1 and Filter2 will only change the proportion of different spectrum, but will not change the information of reconstructed images. Our purpose is to correct the abnormal spatial spectrum in 3DSIM through the filters, and make up for the spike, mutation, depression, and other factors in the spectrum so that the 3DSIM spectrum is close to the ideal and uniform three-

dimensional spatial spectrum. In this process, RL deconvolution or regularization is not used to challenge the fidelity of the image, so our results are highly reliable. For this reason, we added the simulation using the resolution test version as shown in **Figure R3.4 (Supplementary Note 6 in manuscript)**. It can be seen that Open-3DSIM has no visible artifacts, and has achieved high fidelity compared with GT images, which also proves the correctness of the two-step spectrum filtering.

Figure R3.4 | The comparison of GT, WF, and Open-3DSIM using a resolution test image. Scale bar: 2 μm . Scale on z-axis: 41 layers, 0.125 μm per layer.

Secondly, we compare the two-step frequency domain filtering and Weiner-based SIM and find that our method is similar to the Wiener reconstruction expression. There are OTF square terms in the denominator, but we introduce the OTF term to make a flexible transformation, making the spatial attenuation more controllable, not limited to the OTF square attenuation. At the same time, the numerator of two-step frequency domain filtering is the square of Apo, but the possible high-frequency loss will be compensated by Filter2. The same OTF square terms of the Wiener filter and two-step filter prove the correctness of the two OTF multiplication.

$$\text{Weiner-based: } C_{wiener}(k) = C_{ns}(k) \cdot \frac{Apo}{\sum_{n=-2}^2 OTF^2(x,y,z,n)+w}$$

$$\text{Open-3DSIM: } C_{SR_3}(k) = C_{ns}(k) \cdot \frac{Apo^2}{OTF_{notch}^2+(w1+w2) \cdot OTF_{notch}+w1 \cdot w2}$$

(4) Spectrum filtering's comparison:

In **Supplementary Note 3**, we mainly emphasize the improvement of a two-step spectrum filter for reconstruction results. The comparison with other algorithms can be easily seen in **Fig 2a-d** and **Supplementary Note 7-9**. In response to your comments, we added the frequency domain image of **Fig2a** as shown in **Fig R3.5**. It can be seen that under the condition of an extremely low signal-to-noise ratio, the high-frequency component of OMX is missing and the high-frequency spike is prominent, resulting in serious honeycomb artifacts. The high-frequency

peak of SIMnoise is prominent, but the high-frequency center is sunk, resulting in the reduction of proportion between high and low frequency components, resulting in more defocused backgrounds and abnormal spectrum. Through spectrum optimization, Open-3DSIM also maintains a uniform petal-shaped spectrum under extremely low SNR, thus achieving good reconstruction results.

Figure R3.5 | Comparison of reconstruction results and the spectrum between OMX, SIMnoise, and Open-3DSIM. Scale bar: 2 μ m. Scale on z-axis: 13 layers, 0.125 μ m per layer.

Action:

On Manuscript, Page 4, Line 3-4, added the description.

On Manuscript, Page 5, Line 14, added the description.

On Supplementary Note 1, explained the principle of Open-3DSIM in detail;

On Supplementary Note 3, added the reconstruction results and spectrums (embed top right) between different algorithms;

On Supplementary Note 6, added the simulation of resolution test.

Reference

1. Demmerle. J, *et. al.* Strategic and practical guidelines for successful structured illumination microscopy. *Nature Protocal* **12**, 988–1010 (2017).
2. Karras. C, *et. al.* Successful optimization of reconstruction parameters in structured illumination microscopy – a practical guide. *Optical Communications* **436**, 69–75 (2019).
3. Huang. X, *et. al.* Fast, long-term, super-resolution imaging with Hessian structured illumination microscopy. *Nature Biotechnolgy* **36**, 451–459 (2018).
4. Wen. G, *et. al.* High-fidelity structured illumination microscopy by point-spread-function

engineering. Light: Science & Applications **10**, 1-12 (2021).

Decision Letter, first revision:

Dear Peng,

Thank you for submitting your revised manuscript "Open-3DSIM: an Open-source three-dimensional structured illumination microscopy reconstruction platform" (NMETH-BC51095A). It has now been seen by the original referees and their comments are below. The reviewers find that the paper has improved in revision, and therefore we'll be happy in principle to publish it in Nature Methods, pending minor revisions to comply with our editorial and formatting guidelines.

You will see that reviewer 1 has one additional request for your software (to add a progress bar). If this is not straightforward to address in a revision, we do not require it, but we ask that you strongly consider adding it in updated versions of your tool.

TRANSPARENT PEER REVIEW

Nature Methods offers a transparent peer review option for new original research manuscripts submitted from 17th February 2021. We encourage increased transparency in peer review by publishing the reviewer comments, author rebuttal letters and editorial decision letters if the authors agree. Such peer review material is made available as a supplementary peer review file. Please state in the cover letter 'I wish to participate in transparent peer review' if you want to opt in, or 'I do not wish to participate in transparent peer review' if you don't. Failure to state your preference will result in delays in accepting your manuscript for publication.

ORCID

Sincerely,
Rita

Rita Strack, Ph.D.
Senior Editor
Nature Methods

Reviewer #1 (Remarks to the Author):

See attached PDF

[Attached]:

In the revised version of their manuscript entitled "Open-3DSIM: an Open-source three-dimensional structured illumination microscopy reconstruction platform" authors Ruijie Cao et al. address the issues raised in the review process, and in my opinion greatly improve upon their original submission. **In its improved form, I support publication of the manuscript.**

I have focussed mainly on testing the software itself, as here I encountered most problems with the code originally submitted. I am happy to see that the authors have approved on the following points:

- 1) The MATLAB version of their code now runs without error messages or references to non-existing functions, even on different versions of the MATLAB platform. It does not provide a GUI, but I do not see this as a drawback, as a GUI is offered both by the stand-alone and the FIJI integrated version of their software
- 2) There is a stand-alone version of their software that offers two in my opinion important features to end users: It runs without requiring a MATLAB installation and it provides an intuitive GUI to load images and set reconstruction parameters. I found the software easy to install and operate. It does require installing a MATLAB runtime environment (as it is still based on packaged MATLAB code), but this is free of charge and easy to install.
- 3) Lastly, the version integrating with ImageJ / FIJI has greatly been improved. While being implemented as packaged MATLAB code (relying on the runtime environment), the authors greatly improved integration with FIJI: The raw input images are now imported from image stacks opened in FIJI, and the resulting reconstructed data is passed back to FIJI and directly presented as FIJI stacks. This greatly improved interoperability, as now all FIJI processing features (LUT, contrast setting, registration, ...) can be directly used on the images. Also, the new version provides some intermediate results of the parameter estimation feature (as it seems similar to the same feature in fairSIM), which is known to help with debugging challenging datasets.

I have one suggestion that still could be implemented in all three versions of the software, but is mostly a comfort feature: Especially with larger datasets, parameter estimation and reconstruction is time-consuming. Currently, the software does not give any feedback if the parameter search is still in progress, or if for some reason the code has crashed. A progress bar feature or something similar could tell the user that the software is still working.

Clear license statement: I am very happy to see the authors chose a very open license (BSD) for their code. I would recommend adding this license statement not only to the manuscript, but especially to the github repository (the standard way is in form of a license file in the main folder, and typically even at the top of every source file) to clearly indicate the license to others who interact with the code through github and similar repositories.

Reviewer #2 (Remarks to the Author):

The authors have replied all question with many new materials and modified their manuscript according. Now I would like to see the publishing of this paper and promotion of related research in the field.

Reviewer #3 (Remarks to the Author):

I'm okay with the revisions and responses.

Author Rebuttal, first revision:

Response to Reviewers' Comments

We very much appreciate the critical reading of our manuscript and valuable suggestions from the both editor and reviewers. We have carefully reviewed the comments and have revised the manuscript accordingly. To address all the comments, we reorganized the manuscript in the type of research article and underlined significant changes to facilitate the review of the revised manuscript. The responses to the comments are listed one by one as follows (in blue):

To editor:

Thank you for submitting your revised manuscript "Open-3DSIM: an Open-source three-dimensional structured illumination microscopy reconstruction platform" (NMETH-BC51095A). It has now been seen by the original referees and their comments are below. The reviewers find that the paper has improved in revision, and therefore we'll be happy in principle to publish it in Nature Methods, pending minor revisions to comply with our editorial and formatting guidelines.

You will see that reviewer 1 has one additional request for your software (to add a progress bar). If this is not straightforward to address in a revision, we do not require it, but we ask that you strongly consider adding it in updated versions of your tool.

Response: Dear editor, thank you for your support of our manuscript "Open-3DSIM: an Open-source three-dimensional structured illumination microscopy reconstruction platform". We have provided the progress bar for the latest version of Open-3DSIM (v2.1), including MATLAB, Fiji, and Exe versions as shown in **Fig. E1**. We hope these changes will help users to use Open-3DSIM

conveniently.

Figure E1 | The progress bar. We have added the progress bar in (a) MATLAB, (b) Fiji, and (c) Exe version of Open-3DSIM.

Reviewer #1 (Remarks to the Author):

In the revised version of their manuscript entitled “Open-3DSIM: an Open-source three-dimensional structured illumination microscopy reconstruction platform” authors Ruijie Cao et al. address the issues raised in the review process, and in my opinion greatly improve upon their original submission. In its improved form, I support publication of the manuscript.

Response: Dear Reviewer, we thank you very much for your support for Open-3DSIM.

I have focussed mainly on testing the software itself, as here I encountered most problems with the code originally submitted. I am happy to see that the authors have approved on the following points:

1) The MATLAB version of their code now runs without error messages or references to non-existing functions, even on different versions of the MATLAB platform. It does not provide a GUI, but I do not see this as a drawback, as a GUI is offered both by the stand-alone and the FIJI integrated version of their software

2) There is a stand-alone version of their software that offers two in my opinion important features to end users: It runs without requiring a MATLAB installation and it provides an intuitive GUI to load images and set reconstruction parameters. I found the software easy to install and operate. It does require installing a MATLAB runtime environment (as it is still based on packaged MATLAB code), but this is free of charge and easy to install.

3) Lastly, the version integrating with ImageJ / FIJI has greatly been improved. While being implemented as packaged MATLAB code (relying on the runtime environment), the authors greatly improved integration with FIJI: The raw input images are now imported from image stacks opened in FIJI, and the resulting reconstructed data is passed back to FIJI and directly presented as FIJI stacks. This greatly improved interoperability, as now all FIJI processing features (LUT, contrast setting, registration, ...) can be directly used on the images. Also, the new version provides some intermediate results of the parameter estimation feature (as it seems similar to the same feature in fairSIM), which is known to help with debugging challenging datasets.

Response: We thank you very much for your support for our three platforms of Open-3DSIM.

I have one suggestion that still could be implemented in all three versions of the software, but is mostly a comfort feature: Especially with larger datasets, parameter estimation and reconstruction is time-consuming. Currently, the software does not give any feedback if the parameter search is still in progress, or if for some reason the code has crashed. A progress bar feature or something similar could tell the user that the software is still working.

Response: Thank you for your helpful suggestions. We have provided the progress bar for the latest version of Open-3DSIM (v2.1), including MATLAB, Fiji, and Exe versions as shown in **Fig. R1.1**. We hope these changes will help users to use Open-3DSIM conveniently.

Figure R1.1 | The progress bar. We have added the progress bar in (a) MATLAB, (b) Fiji, and (c) Exe version of Open-3DSIM.

Clear license statement: I am very happy to see the authors chose a very open license (BSD) for their code. I would recommend adding this license statement not only to the manuscript, but especially to the github repository (the standard way is in form of a license file in the main folder, and typically even at the top of every source file) to clearly indicate the license to others who interact with the code through github and similar repositories.

Response: Thank you for your helpful suggestions. We have added the Apache license in the main folder of our repository and source file.

Name	Last commit message
Exe_Open_3DSIM	Add files via upload
Java_Open_3DSIM	Add files via upload
MATLAB_Open_3DSIM	Update get_vectormodelOTF.m
License	License
Open_3DSIM.rar	Open3DSIM_platform
README.md	Update README.md

Figure R1.2 | The added license file.

Reviewer #2 (Remarks to the Author):

The authors have replied all question with many new materials and modified their manuscript according. Now I would like to see the publishing of this paper and promotion of related research in the field.

Response: We thank the reviewer very much for your previous suggestions, which greatly improves our manuscript.

Reviewer #3 (Remarks to the Author):

I'm okay with the revisions and responses.

Response: We thank the reviewer very much for your previous suggestions, which greatly improves our manuscript.

Final Decision Letter:

Dear Peng,

I am pleased to inform you that your Brief Communication, "Open-3DSIM: an Open-source three-dimensional structured illumination microscopy reconstruction platform", has now been accepted for publication in Nature Methods. Your paper is tentatively scheduled for publication in our August print issue, and will be published online prior to that. The received and accepted dates will be Dec 1, 2022 and June 12, 2023. This note is intended to let you know what to expect from us over the next month or so, and to let you know where to address any further questions.

Over the next few weeks, your paper will be copyedited to ensure that it conforms to Nature Methods style. Once your paper is typeset, you will receive an email with a link to choose the appropriate publishing options for your paper and our Author Services team will be in touch regarding any additional information that may be required.

Your paper will now be copyedited to ensure that it conforms to Nature Methods style. Once proofs are generated, they will be sent to you electronically and you will be asked to send a corrected version within 24 hours. It is extremely important that you let us know now whether you will be difficult to contact over the next month. If this is the case, we ask that you send us the contact information (email, phone and fax) of someone who will be able to check the proofs and deal with any last-minute problems.

If, when you receive your proof, you cannot meet the deadline, please inform us at rjsproduction@springernature.com immediately.

Once your manuscript is typeset and you have completed the appropriate grant of rights, you will receive a link to your electronic proof via email with a request to make any corrections within 48 hours. If, when you receive your proof, you cannot meet this deadline, please inform us at rjsproduction@springernature.com immediately.

Once your paper has been scheduled for online publication, the Nature press office will be in touch to confirm the details.

Content is published online weekly on Mondays and Thursdays, and the embargo is set at 16:00 London time (GMT)/11:00 am US Eastern time (EST) on the day of publication. If you need to know the exact publication date or when the news embargo will be lifted, please contact our press office after you have submitted your proof corrections. Now is the time to inform your Public Relations or Press Office about your paper, as they might be interested in promoting its publication. This will allow them time to prepare an accurate and satisfactory press release. Include your manuscript tracking number NMETH-BC51095B and the name of the journal, which they will need when they contact our office.

About one week before your paper is published online, we shall be distributing a press release to news organizations worldwide, which may include details of your work. We are happy for your institution or funding agency to prepare its own press release, but it must mention the embargo date and Nature Methods. Our Press Office will contact you closer to the time of publication, but if you or your Press Office have any inquiries in the meantime, please contact press@nature.com.

If you are active on Twitter, please e-mail me your and your coauthors' Twitter handles so that we may tag you when the paper is published.

Please note that *Nature Methods* is a Transformative Journal (TJ). Authors may publish their research with us through the traditional subscription access route or make their paper immediately open access through payment of an article-processing charge (APC). Authors will not be required to make a final decision about access to their article until it has been accepted. [Find out more about Transformative Journals](https://www.springernature.com/gp/open-research/transformative-journals)

Authors may need to take specific actions to achieve [compliance](https://www.springernature.com/gp/open-research/funding/policy-compliance-faqs) with funder and institutional open access mandates. If your research is supported by a funder that requires immediate open access (e.g. according to a

[Plan S principles](https://www.springernature.com/gp/open-research/plan-s-compliance)) then you should select the gold OA route, and we will direct you to the compliant route where possible. For authors selecting the subscription publication route, the journal's standard licensing terms will need to be accepted, including [self-archiving policies](https://www.springernature.com/gp/open-research/policies/journal-policies). Those licensing terms will supersede any other terms that the author or any third party may assert apply to any version of the manuscript.

To assist our authors in disseminating their research to the broader community, our SharedIt initiative provides you with a unique shareable link that will allow anyone (with or without a subscription) to read the published article. Recipients of the link with a subscription will also be able to download and print the PDF. As soon as your article is published, you will receive an automated email with your shareable link.

Please note that you and your coauthors may order reprints and single copies of the issue containing your article through Springer Nature Limited's reprint website, which is located at <http://www.nature.com/reprints/author-reprints.html>. If there are any questions about reprints please send an email to author-reprints@nature.com and someone will assist you.

Best regards,
Rita

Rita Strack, Ph.D.
Senior Editor
Nature Methods